# LoC-Decomp: LLM Autoformalization via Logical Concept Decomposition and Iterative Feedback Correction

**Jiangze Shi**[1], **Zhiwei Zhang**[1]*, **Baoquan Ma**[2], **Shuai Zhao**[1], **Ye Yuan**[1], **Guoren Wang**[1]

[1]School of Computer Science, Beijing Institute of Technology

[2]Lab of UAVs, National Computer System Enginering Research Institute of China

{jzshi,zwzhang,shuaizhao,yuan-ye}@bit.edu.cn;baoquan.ma@buaa.edu.cn;
wanggrbit@126.com;

## Abstract

Autoformalization—the process of converting natural language mathematical statements into machine-verifiable formal code—plays a critical role in ensuring the reliability of mathematical reasoning generated by large language models (LLMs). Recent studies show that LLMs exhibit strong potential in automating this process, producing formal code for systems such as Lean 4, Coq, and Isabelle. Despite prominent advances, existing LLM-based autoformalization methods remain limited: they lack the ability to provide reliable semantic consistency checks to ensure that the formal code accurately preserves the meaning of the original statement. Furthermore, such methods are unable to support iterative improvement through corrective feedback. To address these limitations, we propose Loc-Decomp, a novel framework that integrates an automatic semantic consistency checker and the Lean 4 compiler to iteratively refine LLM-generated formalizations, ensuring both semantic consistency and syntactic correctness. Our approach introduces three key innovations: **(1)** A structured and COT-like formalization template that decomposes complex formalization tasks into modular, foundational components, and systematically assembles them—like building blocks—into a complete formal expression. **(2)** A semantic self-checking mechanism based on a divide-conquer-merge strategy to detect subtle inconsistencies between the formalization and the original statement. **(3)** An iterative feedback-driven refinement loop that leverages both semantic and syntactic error signals to guide the LLM in progressively improving the formal output. By integrating these innovations, Loc-Decomp significantly enhances the accuracy of LLM-driven formalization, reduces reliance on human intervention, and moves closer to truly reliable automated reasoning. Extensive experiments on high-school-level and undergraduate-level datasets demonstrate that our approach achieves a significantly higher formalization success rate compared to baseline methods and state-of-the-art (SOTA) models. On the PutnamBench dataset, for instance, our method attains a success rate of 93.09%, representing an improvement of 18 percentage points over the previous SOTA SFT-based model.

## 1 Introduction

Statement formalization(Weng et al., 2025; Gonthier, 2007; Szegedy, 2020) denotes the process of converting a mathematical statement into a formal language—such as Lean 4, Coq, or Isabelle (de Moura et al., 2015; Bertot & Castéran, 2004; Paulson, 1994)—which constitutes a necessary step for the verification of mathematical reasoning in theorem provers(Zhou et al., 2024). A successful formalization must not only satisfy syntactic correctness, as verified by compiler checks, but also ensure semantic consistency by faithfully preserving the meaning of the original statement. However, this task is widely recognized as highly labor-intensive, owing to the inherent flexibility and ambiguity of natural language, which pose significant challenges to automation (Yang et al., 2024).

---

*Corresponding Author

Recent studies leveraging large language models (LLMs) (Achiam et al., 2023; Team et al., 2023; Liu et al., 2024; Azerbayev et al., 2023b) for autoformalization have shown promising progress and achieved notable results on relatively simple statements. These approaches include both prompt engineering with candidate scoring using general-purpose LLMs and methods based on supervised fine-tuning on domain-specific datasets (Li et al., 2024a; Ying et al., 2024). However, when dealing with more complex mathematical statements—such as those in probability, combinatorics, or geometry (Trinh et al., 2024)— there is still considerable room for improvement. Key challenges include accurately detecting subtle semantic inconsistencies and effectively leveraging these detected inconsistencies to refine the formalization.

To address these two challenges, we proposed an automated LLM-based formalization framework built upon **Lo**gical **C**oncept **Decomp**osition (LoC-Decomp), which integrates a modular formalization template, an automatic semantic consistency check (ASCC) method, and an iterative refinement method within Lean 4. Specifically, we instruct the LLM to generate Lean 4 code conforming to a predefined template that decomposes the formal statement into multiple declaration segments,thereby enabling a semantically complete breakdown. A divide-conquer-merge based back-translation process is then applied to more accurately capture subtle semantics in Lean 4. Subsequently, we perform both segmented and holistic discrepancy detection by prompting the LLM to identify potential semantic inconsistencies within individual segments and the entire statement. The detected discrepancies are then evaluated against predefined criteria by LLM, and rectification suggestions are also provided along with the evaluation procedure. Utilizing these discrepancy descriptions and recommendations as well as compiler error messages, an iterative refinement strategy is implemented to achieve both semantic consistency and syntactic correctness.

As in previous works, we conducted extensive experiments on widely used mathematical datasets, MATH-500(Lightman et al., 2024) and miniF2F(Zheng et al., 2022) for example, to evaluate the effectiveness of our proposed approach. The results indicate that after incorporating the Loc-Decomp Lean 4 template and few-shot examples, the single-round (pass@1) formalization success rate on the miniF2F dataset reached 77.25%. With the further integration of semantic consistency checks, compiler verification, and iterative feedback refinement, the pass rate on miniF2F increased to 90.16%.[1]. Experimental results and code are available at `https://github.com/jzshischolar/auto-Formalization`.

In summary, the main contributions of this paper are as follows:

1. We introduced LoC-Decomp, a COT-like Lean 4 template that breaks down the formalization process into multiple steps, with a theoretical guarantee of expressiveness.

2. We introduced a novel semantic consistency checking method by decomposing the formalization code. This approach enables LLMs to accurately detect semantic inconsistencies between the formalized code and the original problem in complex scenarios.

3. We presented an iterative feedback-based method for semantic and syntactic correction, allowing the LLM to leverage identified semantic inconsistencies from the previous step or compiler-generated syntax errors to iteratively refine the Lean 4 code.

4. We conducted extensive experiments on two widely adopted datasets and evaluated the results using an automatic evaluation metric supplemented by human assessment, which collectively demonstrate the effectiveness of the proposed approach.

## 2 BACKGROUND AND RELATED WORK

**Formal Reasoning:** Recently, a number of studies have emerged that employ formal provers such as Lean 4, Coq, and Isabelle to validate the reasoning processes of LLMs (Yang et al., 2023) (Wang et al., 2024; Li et al., 2024b; Lin et al., 2025a; Alfarano et al., 2024; Huang et al., 2024). As pointed out in (Yang et al., 2024), leveraging rigorous formal provers to provide feedback can effectively mitigate data scarcity and counteract hallucinations. Automated theorem provers represent one of the building blocks of this approach: given a formal proposition, they require the LLM to output a proof process, which is then verified using a formal proof system. BFS-prover (Xin et al.,

---

[1]All these results are under the ASCC-3-MV metric with DeepSeek-V3 as base model, see section 4 for more information.

2025) through an optimized Best-First Search framework enhanced by expert iteration and policy refinement. DeepSeek-prover-v2 (Ren et al., 2025) introduces a cold-start reinforcement learning procedure that integrates informal mathematical reasoning with formal proof steps through a recursive theorem-proving pipeline.

**Autoformalization:** Unlike theorem proving, autoformalization does not generate proofs for theorems; rather, it converts natural language statements into formal specifications (Weng et al., 2025). Autoformalization thus acts as a bridge between informal and formal mathematics.Traditional rule-based autoformalization methods (Pathak, 2024) are limited by their manual design, fragility to unseen constructs, and poor semantic disambiguation. In contrast, methods based on Large Language Models (LLMs) offer greater flexibility and are capable of capturing subtle or rare linguistic patterns that might be overlooked by human experts during rule design. LLM-based autoformalization primarily follows two research directions: fine-tuning on synthetically generated data (Jiang et al., 2025; 2023b;a; Azerbayev et al., 2023a) and prompt-based (in-context learning) approaches. Some reserchers observed that auto-informalization is easier than autoformalization (Wu et al., 2022). This observation has motivated subsequent work that employs LLM-based back-translation to verify the correctness of formalized outputs (Li et al., 2024a). Another line of research, parallel to our work, explores the use of Retrieval-Augmented Generation to improve the formalization abilities of large language models (Liu et al., 2025a). A training based evaluation method is proposed by Lu et al. (2024a) to detect misalignment in formalization, but this method can only provide a numerical score without targeted revision suggestions. For feedback refinement, several studies have utilized compiler error messages to improve the formalization results generated by LLMs(Liu et al., 2025b; Lu et al., 2024b; Zhang et al., 2024). The work of KELPS (Zhang et al., 2025) introduced an intermediate language to facilitate concept decomposition; however, it did not integrate this decomposition idea into semantic consistency checking. And FIMO(Liu et al., 2023) was the first to introduce a mechanism that integrates semantic feedback with syntactic error correction. However, a significant limitation is that their semantic feedback mechanism required manual involvement. In summary, an automated approach that integrates semantic inconsistency feedback with compiler errors to simultaneously achieve semantic consistency and syntactic correctness remains unexplored.

## 3 Loc-Decomp Based Iteratively Refinement Framework

As shown in Figure 1, our proposed autoformalization framework consists of three modules: (1) template-based formal translation (FormalTrans) (2) automatic semantic consistency check and iterative rectification (ASCC-R), and (3) compiler check and iterative rectification (CpC-R).

The core idea of our approach is as follows:

1. Construct a Lean 4 template that guides the LLM to generate formalization code in a Chain-of-Thought-inspired, stepwise manner (corresponding to the FormalTrans module).

2. Perform fine-grained semantic consistency checking through a divide-and-conquer strategy to detect nuanced semantic inconsistencies (corresponding to the ASCC-R module).

3. Employ an iterative rectification method that jointly addresses semantic inconsistencies and compilation errors (corresponding to the joint iteration of the ASCC-R and CpC-R modules).

At the core of our approach lies a **divide-conquer-merge strategy** for semantic consistency checking. By decomposing the formalized code into logical components through our LoC-Decomp template, we enable fine-grained back-translation and verification of each semantic unit against the original problem statement. This component-level decomposition allows for precise identification and localization of semantic inconsistencies, which then serve as targeted feedback for iterative correction. Coupled with compiler error checking, this creates a closed loop where semantic and syntactic issues are alternately detected and rectified through iterative refinement.

Processing a formalization task follows an iterative workflow across three interconnected modules. First, the **FormalTrans** module synthesizes a prompt from the problem statement, task requirements, few-shot examples, and the LoC-Decomp template, then invokes the LLM to generate initial Lean 4 code which adheres to the template. Second and third, the **ASCC-R** and **CpC-R** modules jointly execute alternating semantic and syntactic refinement: ASCC-R detects inconsistencies through back-

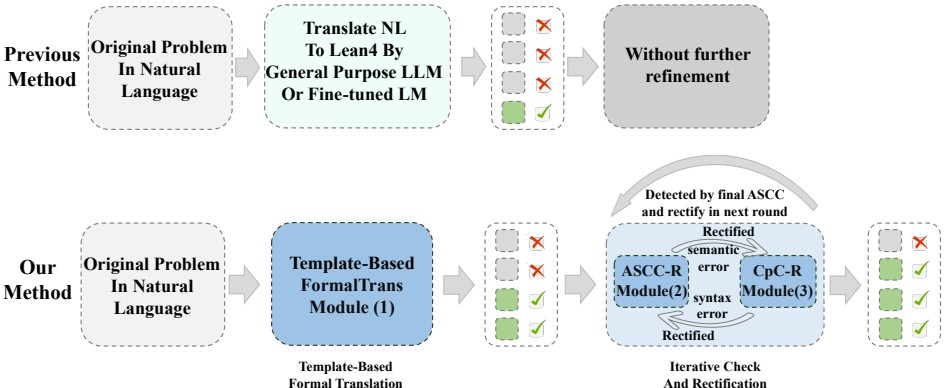

Figure 1: Overview of our Loc-Decomp based iteratively refinement framework (bottom) and an overview of previous method(top). In comparison with previous methods that select one from multiple generated candidates, our approach introduces iterative feedback and refinement, eliminating the need to generate multiple candidates and gains cumulative knowledge from each iteration.

translation and proposes targeted modifications, while CpC-R identifies and resolves compilation errors. Each module operates iteratively within its domain—ASCC-R runs an *inner semantic loop* until semantic consistency is achieved (or the semantic iteration limit $K\text{-}sem$ is reached), while CpC-R runs an *inner syntax loop* until compilation succeeds (or the syntactic iteration limit $K\text{-}syn$ is reached). These inner loops form a *Sem–Syn iteration unit*; multiple such units constitute an *outer loop* that continues until the code passes both checks consecutively or the maximum iteration limit is reached.

## 3.1 FORMAL TRANSLATION MODULE

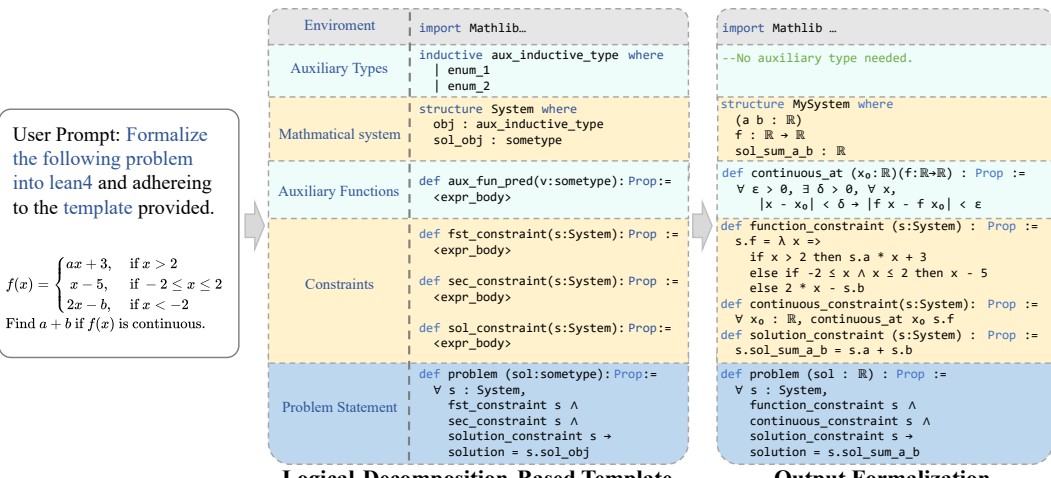

Figure 2: Formalization template and a concrete example. The left is our proposed Template with place holders, and the right is an example for formalizing a problem about the property of a continuous piecewise function. Please note that the example shown in this figure was manually designed to demonstrate the template structure. For examples generated by large language models (LLMs), we refer readers to Appendix F.

The FormalTrans module serves as the entry point of our framework, responsible for transforming mathematical problems in natural language into structured Lean 4 code. Unlike prior approaches that directly formalize problems as Lean 4 theorems, we introduce a novel template-based approach that de-

composes the formalization task into a sequence of semantically meaningful components—including mathematical objects, constraints, and the problem statement[2], as shown in Figure 2. This decomposition mimics a Chain-of-Thought (CoT) reasoning process, guiding the LLM to construct formalizations step-by-step rather than attempting to generate the complete code in a single pass. By breaking down the complex formalization task into manageable subtasks, our template enables LLMs to organize the Lean 4 content in a more structured manner, thereby enhancing performance as demonstrated in our experiments. The formulation of the template is kind of based on Discourse Representation Theory, which is introduced to mathematical formalization by (Ganesalingam, 2013). A detailed explanation for this template is available in A.1.

To make sure the template is fully complied with, a verification method is required. Since this template framework imposes requirements beyond the syntax rules of the Lean 4 compiler, a lightweight parser is implemented to serve as template verifier. If the requirements are not met, it returns corresponding feedback to prompt the LLM to regenerate the code. This process iterates until all conditions are satisfied or the maximum iteration limit is reached. We emphasis that although this template is demonstrated using Lean 4, it can be easily adapted to other theorem provers like Coq or Isabella, a detailed discussion is provided in A.7 and details about the parser and the feedback strategy are available in A.2.

Since such a template imposes additional requirements on top of Lean syntax, the set of all Lean 4 code that satisfies the template forms a proper subset of all Lean 4 code that meets the syntactic requirements. This implies that Lean 4's expressive power may be constrained under this template, as certain formal declarations might not be convertible into the template's structure. To provide a theoretical guarantee for the expressiveness of the template, we propose the following theorem3.1 and provide its proof in the A.4.

**Definition 3.1.** LoC-Decomp counterpart: For a complete Lean 4 proposition string(by complete we mean that this string contains a proposition defined by theorem or lemma and all it's dependencies), we define its LoC-Decomp counterpart as a string that is semantically equivalent to the Lean 4 proposition and satisfies the LoC-Decomp template requirements. Here, satisfying the LoC-Decomp template requirements means that the string can be accepted by our parser that encodes the syntactic rules of the template.

**Theorem 3.1.** *For any complete Lean 4 proposition string, there exists at least one LoC-Decomp counterpart.*

## 3.2 AUTOMATIC SEMANTIC CONSISTENCY CHECK MODULE

To verify semantic consistency between the formalization and the original statement, our ASCC module employs a back-translation component that converts Lean 4 code into natural language while preserving semantic nuances. This back-translated description, presented alongside the formal code, enables the LLM to assess semantic alignment between the natural language problem and its Lean 4 representation, as illustrated in the top half of Figure 3.

Direct translation of complete Lean 4 code into natural language is unreliable for complex mathematical statements, as LLMs struggle to detect subtle semantic logic in a single pass. To address this limitation, we employ a divide-conquer-merge strategy: **(1)** *divide* the Lean 4 code into semantically self-contained segments where each encapsulates complete, independent units of meaning[3]; **(2)** *conquer* by translating each segment independently via the LLM; and **(3)** *merge* by integrating the translated segments into a cohesive natural language description based on their logical relationships. Detailed discussions of the divide and merge mechanisms are provided in Appendix C.

As shown in the bottom half of Figure 3, we leverage the back-translated natural language description to verify semantic consistency between the formalization and the original problem. This comparison allows us to detect discrepancies that may indicate formalization errors. We employ a four-stage strategy: identifying all potential discrepancies, evaluating the significance of each discrepancy,

---

[2]This template applies to both solving and proof-oriented problems. The main text focuses on solving-type problems, but the framework can be adapted to proof-type problems with minor adjustments (see A.5).

[3]The term "semantically self-contained" refers to segments in which all necessary semantic components—such as definitions or premises—are explicitly contained within the segment itself (i.e., the segment can pass the compiler check).

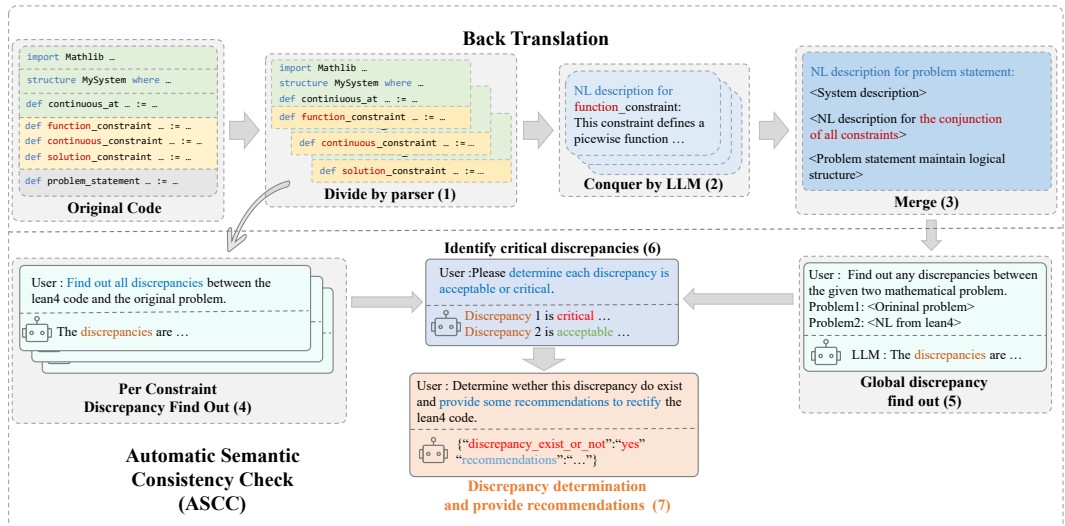

Figure 3: Back translation by divide-conquer-merge strategy and the automatic semantic consistency check.

synthesizing critical discrepancies to determine an overall consistency level, and re-examining each discrepancy in cases not fully consistent—if any discrepancy is confirmed, the output is deemed semantically inconsistent. Specifically, **(4)** segmented and holistic discrepancy detection are employed to identify differences between the formalization and the problem description. The former targets subtle, localized inconsistencies, whereas the latter focuses on overall, structural issues; **(5)** Collective evaluation of identified discrepancies to classify them as critical or acceptable based on predefined criteria, followed by determining an overall consistency level according to all critical discrepancies, the consistency level include: Fully consistent, Consistent without loss of generality, Inconsistent; **(6)** Individual re-evaluation of each discrepancy if the code is not fully consistent; and **(7)** For each confirmed discrepancy, the LLM must provide correction suggestions, and the formalization is judged not fully consistent. If any of discrepancy exist, the formalization is inconsistent. Detailed explanations of each step and judge criteria as well as consistency level definitions are available in Appendix B and prompt used are provided in Appendix I. This consistency-checking mechanism enables us to provide targeted discrepancy information and recommendations, allowing the LLM to rectify the Lean 4 code (as described in following section).

### 3.3 JOINT SYN-SEM ITERATIVE RECTIFICATION

As shown in Figure 4 The iterative rectification procedure is an alternating process of the semantic inconsistency correction and compilation error (i.e., syntax error) correction. Each approach utilizes error feedback to prompt the LLM to reassess its prior output and produce a revised solution. The primary distinction lies in the source of the feedback: semantic error information is derived from semantic consistency checks, while compilation error information is provided by the compiler.In both cases, once error information is obtained, it is appended to the context supplied to the LLM for revision. The updated output then undergoes another round of semantic consistency verification or compilation checking. This iterative cycle continues until the code passes both checks consecutively or the maximum iteration count is reached.

More specifically, our approach involves an iterative process that alternates between semantic and syntactic correction, with the objective of steering the formalization toward semantic consistency and syntactic correctness. For a semantically inconsistent formalization, we first run an inner semantic loop, where semantic corrections are made and re-validated iteratively until success or until reaching the maximum semantic iterations ($K\text{-}sem$). After passing semantic validation, it enters the inner syntactic loop, undergoing compilation checks and syntactic fixes until it compiles successfully or exceeds the allowed syntactic iterations ($K\text{-}syn$). One inner semantic loop followed by one inner syntactic loop forms a Sem–Syn Iter Unit. If the formalization fails to pass

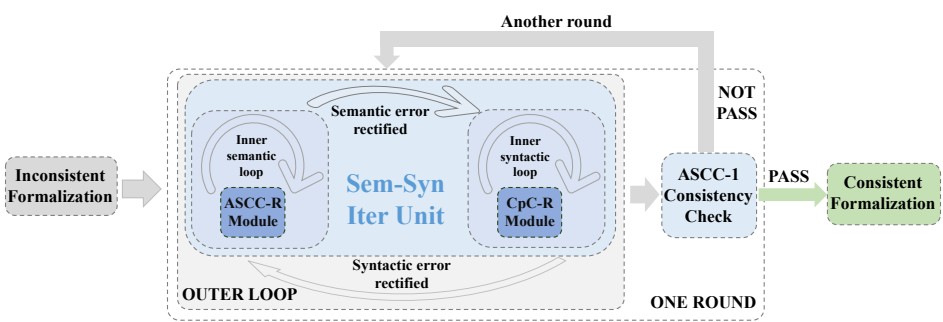

Figure 4: Joint Syn-Sem iterative rectification

both checks consecutively within one unit, it triggers another Sem–Syn Iter Unit; this multi-unit process is the OUTER LOOP, which can run up to $N$ times. After the OUTER LOOP, a final validation for both semantic and syntactic errors is performed—this entire sequence, including at most $N$ outer loops plus the final check, constitutes one ROUND. Failed cases proceed to the next ROUND, with the entire process repeating for up to $M$ rounds. A detailed description of this process is provided in Algorithm 1. In this algorithm, the function ASCC$(\cdot)$ returns the result of the ASCC check, denoted as $ASCCLevel$, which can take one of three values: Fully Consistent, Consistent Without Loss of Generality, or Inconsistent. Here, ASCC-R$(\cdot)$ refers to the ASCC check with rectification, and CpC-R$(\cdot)$ denotes the compiler check with rectification.

---

**Algorithm 1** Iterative check and rectification

---

**Input:** $problem$, $Lean4$;
**Output:** $rectifiedLean4$, $finalASCC$;
 1:  $ASCCLevel \leftarrow ASCC(Lean4)$, $m \leftarrow 0$;
 2:  **while** ($ASCCLevel = Inconsistent$ **or** compiler check not pass) **and** $m < M$ **do**
 3:      **while** ($ASCCLevel = null$ **or** $Inconsistent$) **and** iteration limit not reached **do**
 4:          **while** $ASCCLevel \neq Fullyconsistent$ **and** iteration limit not reached **do**
 5:              $Lean4ToNL \leftarrow$ Informalization($Lean4$);
 6:              $Lean4, ASCCLevel \leftarrow$ ASCC-R($Lean4ToNL, Lean4, problem$);
 7:          **end while**
 8:          **if** $ASCCLevel \neq Inconsistent$ **and** compiler check not pass **then**
 9:              **while** compiler check not pass **and** iteration limit not reached **do**
10:                  $Lean4 \leftarrow$ CpC-R($Lean4$);
11:              **end while**
12:              $ASCCLevel \leftarrow null$;
13:          **end if**
14:      **end while**
15:      $rectifiedLean4 \leftarrow Lean4$, $m \leftarrow m + 1$;
16: **end while**
17: **return** $rectifiedLean4$, $ASCC(rectifiedLean4)$;

---

## 4 Evaluation

### 4.1 Experimental setup

**Datesets:** To evaluate our methods, we employed two widely used public datasets—MATH-500(Lightman et al., 2024) and miniF2F(Zheng et al., 2022) as well as two custom datasets: MATH-ASCC-Eval-150 and MATH-Level5-50.The MATH-500 dataset comprises 500 problems sampled from the MATH-12500(Hendrycks et al., 2021) dataset, covering a variety of problem types and difficulty levels. The miniF2F dataset contains 488 problems sourced from AIME, AMC, and IMO competitions, which is specifically designed for autoformalization by converting solve-type problems into proof-type problems. To evaluate the performance of our method on more complex mathematical prob-

lems, we also adopted PutnamBench(Tsoukalas et al., 2024) and ProofNet(Azerbayev et al., 2023a) as test datasets for further evaluation. PutnamBench and ProofNet contain 522 and 371 undergraduate-level mathematical problems respectively, presenting additional challenges for the autoformalization by LLMs.

The MATH-ASCC-Eval-150 dataset comprises 150 problems sampled from MATH-500 across difficulty levels, each accompanied by Lean 4 formalizations and human annotations (91 positive, 59 negative). The negative cases predominantly contain subtle errors, making this dataset suitable for evaluating automated consistency-checking methods (see Appendix Appendix E for details). In contrast, the MATH-Level5-50 subset consists of 50 randomly selected level-5 problems from MATH-500, designed to assess the formalization capability on challenging problems.

**Models:** Experiments were conducted using three open source LLMs—including DeepSeek-V3, KIMI-K2 and Qwen3-235B(Liu et al., 2024; Yang et al., 2025; Kimi Team et al., 2025)—to validate the effectiveness of the proposed method across different LLMs.

**Baseline:** For a fair comparison with our method, we implemented several baseline methods for three tasks. For the semantic consistency checking task, we introduced two baselines: (1) SC-Baseline, where the LLM is provided with both the original problem and its corresponding Lean 4 code then prompted to assess semantic consistency; and (2) SC-Baseline-BackTrans, which additionally supplies a back translation of the Lean 4 code. For the pass@1 formalization task, we developed (3) Baseline, which employs a basic few-shot prompt to guide the LLM in formalizing the given problem. For the iterative formalization task, we designed (4) Baseline-iter, which implements an iterative method identical to ours but employs a basic semantic checker like the SC-Baseline. To compare our method with the Supervised Fine-Tuning based models, we designed SFT-Baseline by selecting DeepSeek-Prover-V2(Ren et al., 2025) as the base model, and other configurations are exactly the same with the Formal-Baseline. The prompts and detailed configurations for these baseline methods are provided in the Appendix I.

**Metrics:** For the semantic consistency evaluation, we assessed the ASCC method on the MATH-ASCC-Eval-150 dataset as well as the PutnamBench-ASCC-Eval-50 dataset using recall, precision, and F1-score. For the back translationi task, we assess the metric of translation success rate by human evaluation. For the iterative rectification task, the primary evaluation metric was the ASCC-3-MV pass rate—determined through majority voting over three rounds of ASCC—supplemented by human-evaluated pass rates. In addition to ASCC-3-MV, we also report the ASCC-3-SV pass rate—based on single veto over three rounds of ASCC—as a reference metric, due to its high precision but low recall.

**Implementation:** In multi-round iterative testing, we set the maximum iteration numbers as $K\text{-}sem = 2$, $K\text{-}syn = 3$, $N = 5$, $M = 3$. Across all evaluations, the temperature parameter was set to 0.7 for ASCC and 0.3 for all other requests, consistently applied to all LLMs. And all the baseline methods are conducted with DeepSeek-V3.

## 4.2 Experimental Results

**ASCC-3 Evaluation.** We first evaluate the alignment between the ASCC and human judgment criteria for assessing formalization semantic consistency. As shown in Figure 5, on the MATH-ASCC-Eval-150 dataset, the ASCC-3-MV metric achieves a precision of 0.90, a recall of 0.82, and a F1-score of 0.86. Compared to the baseline, precision shows significant improvement—a key focus, as it reflects the method's capability to accurately identify errors. Although precision slightly decreases relative to ASCC-3-SV, recall remains at a reasonably high level. We therefore conclude that ASCC-3-MV aligns most closely with human evaluation criteria. Owing to its strong performance in detecting negative instances, we propose ASCC-3-MV as a standard for evaluating the performance of our methods, while still providing ASCC-3-SV as a more stringent reference.

**Pass@1 with the template and few-shot examples.** As an ablation study, we evaluate our approach without the iterative rectification process to assess the contribution of LoC-Decomp template to the overall pass rate. The results in Table 1 indicate that even without iterative rectification, our method still achieves a relatively high pass rate. We attribute this performance to the chain-of-thought-style template and the use of few-shot examples based on classification. This comparison confirms that

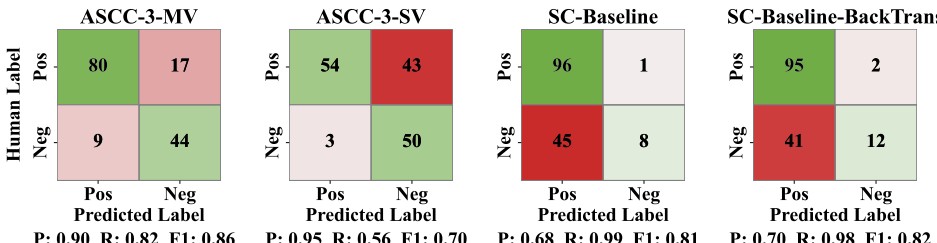

Figure 5: Confusion matrix of the ASCC-3-MV, ASCC-3-SV, SC-Baseline and SC-Baseline-BackTrans (basline with back translation) evaluation result.

our iterative rectification strategy results in a clear performance improvement, with increases in all evaluated items.

Table 1: Results without iterative rectification

| Items | DeepSeek-V3 | | KIMI-K2 | | Qwen3-235B | |
|---|---|---|---|---|---|---|
| | MV | SV | MV | SV | MV | SV |
| MATH-500 | 63.20 | 44.80 | 60.20 | 43.40 | 65.40 | 45.60 |
| miniF2F | 77.25 | 54.30 | 78.57 | 60.29 | 76.90 | 60.29 |

**Joint Syn-Sem iterative rectification evaluation.** The results of the iterative rectification method are summarized in Table 2, where MV denotes the ASCC-3-MV pass rate, and SV denotes the ASCC-3-SV pass rate. The best-performing method is underlined. After three iterations, our approach achieved a ASCC-3-MV pass rate of 84.00% on MATH-500 and 90.16% on miniF2F when combined with DeepSeek-V3, representing the highest performance among all three models evaluated. The result with iterative correction demonstrates a significant improvement over pass@1, which confirms the effectiveness of our proposed method[4].

Table 2: Iterative rectification round-3 results

| Items | DeepSeek-V3 | | KIMI-K2 | | Qwen3-235B | |
|---|---|---|---|---|---|---|
| | MV | SV | MV | SV | MV | SV |
| MATH-500 | 84.00 | 61.60 | 77.80 | 55.60 | 75.60 | 57.80 |
| miniF2F | 90.16 | 67.00 | 91.60 | 72.90 | 80.25 | 61.07 |

**Human evaluation.** As a supplement to the ASCC-3-MV and ASCC-3-SV metrics, we performed a human evaluation on the MATH-Level5-50 dataset. For comparison, three baseline methods—as outlined in Section 4.1—were implemented. The results indicate that, using only template guidance and few-shot examples, our approach achieves a formalization pass rate that exceeds the baseline by 18% under human evaluation. After three rounds of iterative refinement, the pass rate further increased to 84%, surpassing the baseline by 30%. Even when compared to models fine-tuned with expert iteration, our method achieves a higher pass rate. The results are summarized in Table 3, where "Ours-no-iter" refers to our LoC-Decomp method without iterative rectification, and "Ours-iter" denotes the version with iterative rectification. A case study illustrating the rectification process is provided in Appendix F.

**Evaluation on undergraduate-level problems.** For undergraduate-level mathematical problems, our experimental results indicate that under the ASCC-3-MV evaluation standard, our method still demonstrates a notably high accuracy rate, further evidencing the universality and generalizability of our approach. See Table 4 and Table 5 for the results.

---

[4]Additional experimental results about the iterative process are available in Appendix G

Table 3: Human evaluation on MATH-Level5-50

| Items | Ours-no-iter | Ours-iter | Baseline | Baseline-iter | SFT-Baseline |
|---|---|---|---|---|---|
| pass rate | 70.00 | 84.00 | 52.00 | 54.00 | 66.00 |

Table 4: Results without iterative rectification on PutnamBench and ProofNet

| Items | DeepSeek-V3 | | KIMI-K2 | |
|---|---|---|---|---|
| | MV | SV | MV | SV |
| PutnamBench | 48.37 | 31.71 | 53.66 | 35.16 |
| ProofNet | 38.42 | 27.79 | 43.05 | 29.16 |

Table 5: Iterative rectification round-3 results on PutnamBench and ProofNet

| Items | DeepSeek-V3 | | KIMI-K2 | |
|---|---|---|---|---|
| | MV | SV | MV | SV |
| PutnamBench | 81.50 | 58.33 | 72.36 | 48.78 |
| ProofNet | 67.03 | 48.50 | 70.57 | 52.59 |

**Comparative Analysis.** To compare our method with existing approaches, we selected DeepSeek-Prover-V2-671B, Goedel-Formalizor-V2-32B(Lin et al., 2025b), and Kimina-Autoformalizer-7B for evaluation. To ensure a fair comparison, we adopted the conventional LLM-as-a-judge evaluation framework. The results are summarized in the Table 6.

Table 6: Performance comparison of different theorem provers

| Items | Ours | DeepSeek-Prover | Goedel-Formalizor | Kimina-AutoFormalizor |
|---|---|---|---|---|
| MATH-500 | 96.60 | 84.60 | 90.20 | 63.40 |
| mini-F2F | 97.54 | 87.18 | 93.28 | 87.23 |
| PutnamBench | 93.09 | 75.41 | 78.66 | 61.99 |
| ProofNet | 73.57 | 83.10 | 71.39 | 61.31 |

A notable observation is that our method's performance on the ProofNet dataset is significantly lower than that of DeepSeek-Prover-V2. Analysis reveals that the compilation check pass rate of our method on ProofNet is only 74%, considerably lower than the over 95% pass rate achieved on other benchmark datasets. Manual inspection indicates that most compilation failures are due to unresolved type class instances (e.g., "failed to synthesize" errors). We hypothesize that this issue is related to ProofNet's heavy reliance on advanced Mathlib usage patterns. Since the current method does not incorporate a retrieval-augmented generation (RAG) mechanism, the large language model struggles to accurately retrieve and incorporate relevant Mathlib definitions, thereby compromising compilation success. It should be noted that this limitation is not an inherent flaw of the method itself, but rather an orthogonal challenge that can be effectively addressed by integrating RAG-based extensions.

## 5 LIMITATION

Human evaluation in this study was conducted on a dataset of limited scale, a constraint commonly encountered in this field due to the labor-intensive nature of such evaluations. While this is consistent with the practices of related works that face similar scalability challenges, expanding the size of the human-evaluated dataset in future research could further strengthen the reliability and generalizability of our proposed approach.

## THE USE OF LARGE LANGUAGE MODELS

This study utilized Large Language Models (LLMs) in two distinct roles:

1. Language Polishing and Editing: LLMs were employed as an assistive tool to enhance the language quality and clarity of the manuscript. The initial draft was entirely authored by the researcher, after which the LLM provided suggestions to improve sentence fluency, grammar, and academic tone. All conceptual contributions, arguments, and factual assertions originated from the author. The final manuscript was thoroughly reviewed and approved by the author, who takes full responsibility for its content.

2. LLM as a Research Subject: A key aspect of this research involves the evaluation of LLM capabilities. The outputs generated by the model were systematically analyzed as a primary focus of the study. The corresponding methodology is elaborated in the main text.

## REPRODUCIBILITY STATEMENT

To ensure the reproducibility of our work, we have made our complete codebase, along with the raw experimental results and detailed instructions for setting up the environment, publicly available. All materials have been submitted to an anonymous repository for blind peer review and will be retained upon publication.

## ACKNOWLEDGMENTS

This research was funded by the National Key Research and Development Program of China (Grant No. 2024YFE0209000), the NSFC(Grant No. U23B2019, Grant No. U2441237).

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

## APPENDIX A    DETAILED DISCUSSION ABOUT FORMALIZATION TEMPLATE

### A.1    TEMPLATE DETAILS

The LoC-Decomp template departs from prior approaches that formulate Lean 4 formalizations as `theorem` declarations with `sorry` placeholders. Instead, we represent the original statement as a predicate named `problem_statement` that builds upon a series of previously declared definitions. This design enables LLMs to organize Lean 4 content in a structured, step-by-step manner, thereby enhancing formalization quality.

The template is grounded in a key conceptual insight: any mathematical problem investigates the properties of a *mathematical system*—a collection of objects along with constraints that govern their properties and relationships. We represent such a system using a Lean 4 `structure` named `MySystem`, which encapsulates all relevant mathematical objects, along with a series of predicates that describe the constraints and properties of this structure. The problem itself is then formulated as a predicate that examines a specific property of this system.

Take solve-type problems for example, the template comprises the following six sequential steps:

**1. Environment Declaration.** A fixed, predefined setup imports basic Mathlib dependencies, opens namespaces, and defines some notations.

```
1   import Mathlib
2   -->>declare_enviroment
3   open Real InnerProductGeometry Matrix Topology Filter ENNReal
    Polynomial Classical Complex
4   notation "ℝ^"n => EuclideanSpace ℝ (Fin n)
5   notation "𝕄[" m "," n "]" => Matrix (Fin m) (Fin n) ℝ
6   notation "⟪" x ", " y "⟫" => @inner ℝ _ _ x y
7   variable {V : Type} [NormedAddCommGroup V] [InnerProductSpace ℝ V]
8   noncomputable section
9
10  \-Lean4 Code Here-\
11
12  end
```

Figure A-6: The environment defined in the template. All Lean 4 code resides within this environment, so in what follows, we will simply omit it.

**2. Auxiliary Types Declaration.** The LLM may declare custom types (e.g., structures, inductive types) to model complex concepts, leveraging Lean 4's dependent type system without unnecessary restrictions. Since Lean 4's expressiveness stems from its rich type system, this step imposes minimal requirements—allowing any type declaration that facilitates formalization while preserving the template's overall structure.

**3. Mathematical System Declaration.** A `structure` named `MySystem` is declared to abstractly model the problem, containing all mathematical objects (e.g., functions, equations, geometric shapes, groups) along with their types. This step serves as the foundation for formalization, requiring the LLM to identify and declare every relevant object—from concrete entities like sequences and matrices to abstract structures like groups and topologies—thereby creating a comprehensive representation of the problem's mathematical context.

**4. Auxiliary Functions/Predicates Declaration.** Helper functions or predicates are defined to simplify subsequent constraint expressions. These auxiliary definitions capture recurring patterns or complex relationships, promoting modularity and readability in the formalization.

**5. Constraints Declaration.** The LLM declares predicates that define the logical constraints and relationships among objects in `MySystem`. Each constraint predicate—named with the `_constraint` suffix for clarity—specifies properties or connections that mathematical objects must satisfy, thereby encoding the problem's logical structure in a verifiable form.

**6. Problem Statement Declaration.** The overall problem is formalized as a predicate `problem_statement` over a free variable `solution`. Unlike prior approaches using `theorem` declarations with `sorry` placeholders, we reformulate solving-type problems as propositional functions: for any instance of `MySystem`, the conjunction of all constraints implies a proposition about the `solution` variable. This design establishes a direct formal correspondence between the Lean 4 representation and the original problem's semantics, where the predicate holds precisely when the provided solution is valid.

## A.2 DETAILS ABOUT THE PARSER AND THE RECTIFICATION STRATEGY FOR TEMPLATE COMPLIANCE

The parser we have implemented will parse Lean 4 code according to the template. In the auxiliary type declaration section, no additional requirements will be imposed, ensuring the Lean 4 type system remains fully intact. The system declaration section requires that the Lean 4 code must define a `structure` named `MySystem` using the `structure` keyword. The auxiliary function declaration section mandates that functions must be defined with the `def` keyword, and parameters are strictly prohibited from being of the `MySystem` type (to emphasize the auxiliary nature of these functions). The constraint declaration section requires definitions with the `def` keyword, and parameters must strictly be of the `MySystem` type (highlighting constraints on the system). The problem statement section requires definitions with the `def` keyword and must be an implication expression under universal quantification like '∀ sys :  MySystem,...', where the antecedent consists of all constraint conditions, and the consequent is a predicate term related to the free variable solution (or `proof_goal` for proof-type problems).

If the generated Lean 4 code does not conform to the template, a rectification mechanism is activated. This mechanism provides error feedback from the parser and prompts the LLM to revise the Lean 4 code by appending the error information to the context. Whenever new Lean 4 code is generated, it undergoes parsing and rectification—regardless of the current stage, whether it is formalization, semantic rectification, or syntactic rectification. Some parser error information are as follows:

- Error occurred, constraints must be declared by using `def`.
- Error: in the step of `declare_mathematical_system` you can only declare one structure named `MySystem`.
- Error: in the step of `declare_the_problem_statement` you can only declare one predicate named `problem_statement`.
- Error, the antecedents in the problem statement's implication must contain all the constraints declared in the constraint declaration section.
- ...

## A.3 EXPLANATION FOR NONCOMPUTABLE SECTION

Within the domain of high school mathematics, natural language mathematical descriptions (i.e., classical mathematics) are typically grounded in first-order predicate logic—a framework that differs significantly from the dependent type theory underpinning Lean 4. A key practical distinction lies in the treatment of function domains: classical mathematical definitions often implicitly specify the domain through contextual cues and the expression of the function itself, whereas Lean 4's type system requires explicit and precise domain declarations.

This difference considerably complicates the formalization of classical mathematical texts. Since the domain is frequently left implicit in the original discourse, translating such content into Lean 4 requires inferring the domain—a process that can be non-trivial for more complex functions and is sometimes the explicit goal of certain mathematical exercises. However, this act of deduction may introduce semantic discrepancies if not carefully aligned with the original intent.

In the present work (as opposed to prior approaches), we employ a strategy commonly adopted in the formalization of classical mathematics: placing relevant declarations within a noncomputable section. This allows us to avoid prematurely imposing constructive or computational constraints—such as enforcing specific function domains—unless they are directly stated in the original text. The approach maintains mathematical rigor, as domain constraints remain logically inherent in the definitions and can be formally derived and verified during the formalization process.

It is important to emphasize that this method does not circumvent the core challenges of formalization; rather, by utilizing feedback from Lean 4's type system (provided to the LLM during interaction), we can iteratively and efficiently render these implicit elements explicit while respecting the original mathematical meaning.

## A.4 Proof of Theorem 3.1

Here, we complete the proof of Theorem 3.1 by providing a method to convert any complete Lean 4 proposition into a LoC-Decomp template and demonstrating that the result generated by this method is equivalent to the original Lean 4 proposition.

Without loss of generality, the assumption here is that the proposition to be converted is defined by `theorem` and named as `theorem_to_convert`, along with a series of dependency declarations existing in the context. And the template we selected here is the proof type template. The conversion process involves placing all the aforementioned dependency declarations into the auxiliary type section. In the mathematical system declaration section, the MySystem structure contains only one object: `proof_goal`. We then need to transform the `theorem_to_convert` into a proposition and name it as `theorem_converted_prop`, which can be done manually or simply call `#check theorem_to_convert` to get the corresponding type that is exactly the proposition of the theorem. The constraint declaration section includes only one constraint, `proof_goal_constraint`, which states that `sys.proof_goal = theorem_converted_prop`. In the problem statement declaration section, the implication premise contains solely `proof_goal_constraint`, and the implication conclusion is `sys.proof_goal`.

Once such a conversion is performed, the resulting output will conform to our template and can be accepted by the designated parser. Then we have to prove that the `problem_statement` is logically equivalent to the `theorem_converted_prop`, and we prove this by Lean 4 itself. For the sake of universality, we adopted a generic sorry placeholder to express a proposition. Since our proof operates at the meta level, this proposition can be replaced with any proposition, and our proof can pass the verification of the Lean 4 compiler. See code 1 for detailed information.

Code 1: Convert a theorem to it's Loc-Decomp counterpart and prove the equivalence

```
-->>declare_auxiliary_types
--All the dependent declarations should be declared here.

-->>declare_mathematical_system
structure MySystem where
  proof_goal : Prop

-->>declare_auxiliary_functions_or_predicates
def theorem_converted_prop : Prop := sorry
--The sorry place holder can be replaced by any proposition.

-->>declare_constraints
def proof_goal_constraint (sys:MySystem) : Prop :=
  sys.proof_goal = theorem_converted_prop

-->>declare_the_problem_statement
def problem_statement : Prop :=
   ∀ sys : MySystem,
    proof_goal_constraint sys →
    sys.proof_goal

theorem equivalence : problem_statement ↔ theorem_converted_prop := by
    constructor
    ·
        intro h
        unfold problem_statement at h
        let sys : MySystem := ⟨ theorem_converted_prop⟩
        have constraint : proof_goal_constraint sys := by
            unfold proof_goal_constraint
            rfl
        have h_sys_proof_goal : sys.proof_goal := h sys constraint
        exact h_sys_proof_goal
    ·
        intro h sys h_constraint
        rw [h_constraint]
        exact h
```

Although this conversion is not elegant, it theoretically ensures that our template does not impair Lean 4's expressive power in any way.

A.5   MINOR MODIFICATION FOR PROOF TYPE PROBLEMS

When handling proof-type problems, the sol_object is replaced by a proof_goal of type Prop, which corresponds to the proof objective in the original problem. Similarly, the solution_constraint is replaced by proof_goal_constraint, which defines the proof goal concretely. The free variable solution is removed from the problem_statement, as proof-type problems do not require any free variables. In this context, the consequence in the problem statement corresponds to sys.proof_goal for all problems. With these minor adjustments, the template becomes suitable for proof type problems, and our parser remains effective in detecting any violations of the template. The framework only requires a preliminary check of the problem type to determine which template to use. See figure 6 for an example.

Figure A-7: Converting the example formalization into proof type

Figure A-8: Template suitable for Isabella and Coq example

### A.6 DISCUSSION ON FEW-SHOT EXAMPLES

In this work, we adopt a few-shot in-context learning approach with a set of manually curated examples. Our collection comprises 38 Lean 4 codes that conform to the formalization template and cover a variety of problem domains—including functions, probability, combinatorics, and geometry. These examples are drawn from the MATH-12500 dataset (Hendrycks et al., 2021), with only  blank overlapping with the MATH-500 dataset. For each problem, we first determine its domain and then retrieve approximately 8 to 16 relevant examples from that domain to use as few-shot demonstrations. In our baseline methods, the same set of few-shot examples is used, with the only modification being that the Lean 4 code is transformed to a theorem with sorry.

### A.7 EXAMPLES FOR ISABELLA AND COQ

Our proposed framework is not only compatible with Lean 4, but can also be adapted to other theorem provers such as Isabelle and Coq—given the provision of a suitable parser and minor prompt modifications. This flexibility stems from the template-based design of our framework, which can be applied to any proof assistant that supports a type system, the definition of structures (such as 'record' in Coq), as well as predicates and functions.

For Coq, which—like Lean 4—supports dependent type theory, the adaptation process is relatively straightforward. In the case of Isabelle, which uses a simple type system, certain expressive features may be limited; however, these remain sufficient for handling high-school-level mathematical problems. We illustrate the adaptation of our template to both Isabelle and Coq using a simple example: the same piecewise function problem discussed in the main text.

## APPENDIX B    ASCC WORKFLOW DETAILS

The procedure consists of the following steps:

First, we supply the LLM with segmented segments of the Lean 4 code along with their corresponding natural language translations. The model is then prompted to identify any discrepancies between auxiliary functions or constraints and the original problem, under the assumption that all other components are correct.

Second, the complete Lean 4 code and its full natural language description are provided, and the LLM is instructed to detect any potential inconsistencies between the two.

Third, all identified discrepancies are compiled and presented to the LLM, along with the original problem and the Lean 4 code. The model is then asked to assess whether each discrepancy is critical or acceptable based on predefined criteria.

Fourth, the LLM is required to evaluate the overall consistency level by considering all critical discrepancies that have been identified.

Finally, if the code is deemed not fully consistent, each individual discrepancy is isolated and reevaluated by the LLM to confirm its validity. For each confirmed discrepancy, the model must provide specific recommendations to amend the Lean 4 code. The formalization is considered inconsistent if any discrepancy is judged to be genuine.

### B.1    ASCC JUDGE CRITERIA

**Criterion 1:  Object Omission**
Not all mathematical objects in problem original are reflected in the Lean 4 code or their types mismatch. As long as the objects are correctly reflected, some redundancy in the Lean 4 code are acceptable.

**Criterion 2:  Semantic Alteration**
The definitions, properties, or relationships of any mathematical object from problem original are not exactly preserved in the Lean 4 code. Some redundancy is permitted as long as the core semantics are constrained correctly.

**Criterion 3:  Constraint Incompleteness**
The constraints expressed in problem formal fail to comprehensively represent all explicit and implicit constraints present in problem original.

**Criterion 4:  Over Simplification**
The Lean 4 code conducts concrete computation or derivation that simplifies the original problem, leading to a semantic inconsistency.

### B.2    ASCC CONSISTENCY LEVEL DEFINITIONS

**Consistency levels 1:  Fully consistent**
Fully consistent by the final criterion.

**Consistency levels 2:  Consistent without loss of generality**
Consistent without loss of generality: The Lean 4 code formalizes the problem by analyzing representative cases. In each of these cases, the final criterion are fully satisfied, and the general case follows through straightforward deduction.Or the formalization rely on equivalent conversions, such as transforming canonical equations into general form. In such cases, the formulation can be regarded as consistent without loss of generality.

**Consistency levels 3:  Inconsistent**
Any criterion breaked can lead to this, as long as it is not Consistent without loss of generality.

## APPENDIX C    DIVIDE AND MERGE DETAILS IN BACK TRANSLATION

**Divide:** We adopted a hierarchical approach to decompose the Lean 4 code into multiple self-contained segments. The system segment includes all auxiliary types and the `MySystem` `structure`; the auxiliary function segments comprise each auxiliary function along with its dependencies and the system segment; the constraint segments encapsulate each constraint together with its related auxiliary functions and the system segment. The problem statement itself is not processed separately, as the semantic meaning of the problem statement can be fully constructed from the segments described above. The divide strategy simplifies the complex back translation task into several simpler subtasks. Ensures that each segment can be accurately translated in isolation without relying on outer contextual information from other parts of the code, thereby reducing ambiguity and errors during the LLM processing phase.

**Merge:** After translating all segments, the natural language description of the Lean 4 code could, in principle, be obtained by simply concatenating them and adding a statement of their logical relationships. However, such a direct approach would result in a tediously long output, which could distract the LLM during the subsequent semantic check step. Instead, we perform a conjunctive combination of the constraints in natural language. This is achieved by sequentially merging pairs of constraints and instructing the LLM to restate them cohesively, leveraging the fact that their logical relationship in the original problem statement is simply a conjunction. The final translation of the Lean 4 code consists of three core components: a system description, a auxiliary functions description, and a constraints description, which are integrated with a description of their logical relationships.

## APPENDIX D  HUMAN EVALUATION CRITERIA FOR SEMANTIC CONSISTENCY

The detailed criteria for manual inspection are as follows:

1, Whether the formalized problem discusses the same mathematical objects as the original problem (redundant auxiliary objects are allowed, and individual objects may be expressed as sets);

2, Whether the logical constraints of the original problem are correctly expressed (under the current constraints, all object properties and relationships are consistent with the original problem); if certain constraint expressions are not generally correct but are correct in the current context, they are still considered valid;

3, Whether the truth value of the formalized problem statement corresponds exactly to that of the original problem—that is, for a given set of concrete objects (including a solution in the case of a solve-type problem), the formal statement holds true if and only if the original problem is true;

4, We impose no additional requirements on the form of the solution (e.g., some problems require solution in degrees, but solution in radians are also considered correct as long as they are practically equivalent). This is because Lean 4 has limited support for symbolic computation, making it difficult to satisfy certain formal requirements (e.g., requiring expressions to be in simplest form).

5, Consistent without loss of generality: The Lean 4 code formalizes the problem by analyzing representative cases. In each such case, the final criterion is fully satisfied, and the general case follows through straightforward deduction. Alternatively, the formalization may rely on equivalent conversions—such as transforming canonical equations into general form—or the expression may hold only in the current specific context, even if not generally valid. In these situations, the formulation can be regarded as consistent without loss of generality.

The Lean4 code only expresses the horizontal asymptote in the positive infinity direction and overlooks the horizontal asymptote at negative infinity. However, in this problem, the horizontal asymptotes at negative infinity and positive infinity are the same, so we consider this to be a consistent representation without loss of generality.

**Problem:**

The graph of $f(x) = \frac{2x}{x^2 - 5x - 14}$ has vertical asymptotes $x = a$ and $x = b$, and horizontal asymptote $y = c$. Find $a + b + c$.

```
1   -->>declare_auxiliary_types
2   -- No auxiliary types needed for this problem
3   -->>declare_mathematical_system
4   structure MySystem where
5     f : ℝ → ℝ
6     a : ℝ
7     b : ℝ
8     c : ℝ
9     sol_sum : ℝ
10  -->>declare_auxiliary_functions_or_predicates
11  def has_vertical_asymptote_at (f : ℝ → ℝ) (x₀ : ℝ) : Prop :=
12    ∀ (L : ℝ), ∃ (ε : ℝ), ε > 0 ∧
13    (∀ (x : ℝ), x - x₀ ∧ abs (x - x₀) < ε → abs (f x) > L)
14  def has_horizontal_asymptote_at (f : ℝ → ℝ) (y₀ : ℝ) : Prop :=
15    ∀ (ε : ℝ), ε > 0 → ∃ (M : ℝ), ∀ (x : ℝ), x > M → abs (f x - y₀) < ε
16  -->>declare_constraints
17  def function_expression_constraint (sys : MySystem) : Prop :=
18    sys.f = λ x => (2 * x) / (x^2 - 5 * x - 14)
19  def vertical_asymptotes_constraint (sys : MySystem) : Prop :=
20    has_vertical_asymptote_at sys.f sys.a ∧
21    has_vertical_asymptote_at sys.f sys.b ∧
22    sys.a - sys.b
23  def horizontal_asymptote_constraint (sys : MySystem) : Prop :=
24    has_horizontal_asymptote_at sys.f sys.c
25  def solution_constraint (sys : MySystem) : Prop :=
26    sys.sol_sum = sys.a + sys.b + sys.c
27  -->>declare_the_problem_statement
28  def problem_statement (solution : ℝ) : Prop :=
29    ∀ sys : MySystem,
30    function_expression_constraint sys ∧
31    vertical_asymptotes_constraint sys ∧
32    horizontal_asymptote_constraint sys ∧
33    solution_constraint sys →
34    solution = sys.sol_sum
```

Figure D-9: Case study for consistency without loss of generality.

## APPENDIX E  MATH-ASCC-EVAL-150 DATASET EXPLANATION IN DETAIL

We construct the MATH-ASCC-EVAL-150 dataset through the following steps:

1. Using the method proposed in Section3.1 with DeepSeek-V3, we perform a *pass@1* NL2Lean4 formalization procedure. The generated Lean 4 code is then evaluated by ASCC-1, and each instance is categorized based on (*level*, ASCC-1 result) pairs.

2. From the above outcomes, we randomly sample the following from each category:
   - 10 instances of (level 1, ASCC-1 pass)
   - 10 instances of (level 1, ASCC-1 not pass)
   - 10 instances of (level 2, ASCC-1 pass)
   - 10 instances of (level 2, ASCC-1 not pass)
   - 10 instances of (level 3, ASCC-1 pass)
   - 10 instances of (level 3, ASCC-1 not pass)
   - 20 instances of (level 4, ASCC-1 pass)
   - 20 instances of (level 4, ASCC-1 not pass)
   - 25 instances of (level 5, ASCC-1 pass)
   - 25 instances of (level 5, ASCC-1 not pass)

3. We conduct a human evaluation based on predefined criteria to determine the ground-truth consistency between each problem and its corresponding Lean 4 code.

4. After human evaluation, the final dataset consists of **97 positive cases** and **53 negative cases**.

In MATH-ASCC-Eval-150, the negative cases contain subtle semantic inconsistency, which are hard to detect. Some cases are as follows:

**Case studies for nuanced semantic inconsistencies in MATH-ASCC-Eval-150:**

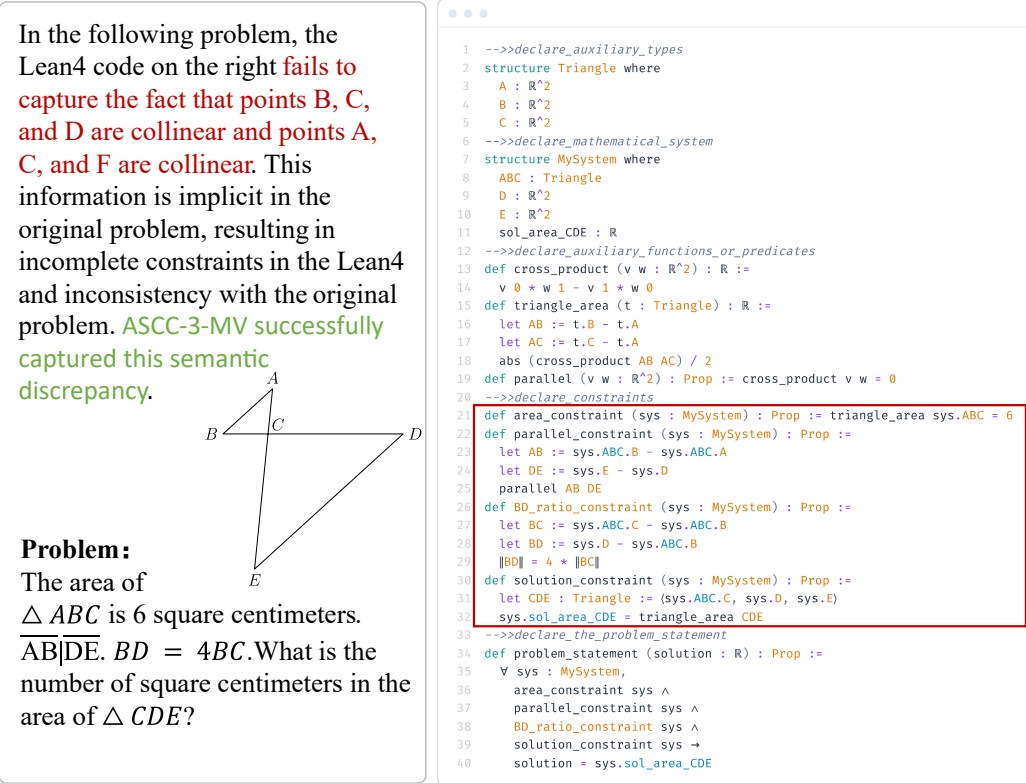

In the following problem, the Lean4 code on the right fails to capture the fact that points B, C, and D are collinear and points A, C, and F are collinear. This information is implicit in the original problem, resulting in incomplete constraints in the Lean4 and inconsistency with the original problem. ASCC-3-MV successfully captured this semantic discrepancy.

**Problem:**
The area of $\triangle ABC$ is 6 square centimeters. $\overline{AB}|\overline{DE}$. $BD = 4BC$. What is the number of square centimeters in the area of $\triangle CDE$?

```
 1  -->>declare_auxiliary_types
 2  structure Triangle where
 3    A : ℝ^2
 4    B : ℝ^2
 5    C : ℝ^2
 6  -->>declare_mathematical_system
 7  structure MySystem where
 8    ABC : Triangle
 9    D : ℝ^2
10    E : ℝ^2
11    sol_area_CDE : ℝ
12  -->>declare_auxiliary_functions_or_predicates
13  def cross_product (v w : ℝ^2) : ℝ :=
14    v 0 * w 1 - v 1 * w 0
15  def triangle_area (t : Triangle) : ℝ :=
16    let AB := t.B - t.A
17    let AC := t.C - t.A
18    abs (cross_product AB AC) / 2
19  def parallel (v w : ℝ^2) : Prop := cross_product v w = 0
20  -->>declare_constraints
21  def area_constraint (sys : MySystem) : Prop := triangle_area sys.ABC = 6
22  def parallel_constraint (sys : MySystem) : Prop :=
23    let AB := sys.ABC.B - sys.ABC.A
24    let DE := sys.E - sys.D
25    parallel AB DE
26  def BD_ratio_constraint (sys : MySystem) : Prop :=
27    let BC := sys.ABC.C - sys.ABC.B
28    let BD := sys.D - sys.ABC.B
29    ‖BD‖ = 4 * ‖BC‖
30  def solution_constraint (sys : MySystem) : Prop :=
31    let CDE : Triangle := (sys.ABC.C, sys.D, sys.E)
32    sys.sol_area_CDE = triangle_area CDE
33  -->>declare_the_problem_statement
34  def problem_statement (solution : ℝ) : Prop :=
35    ∀ sys : MySystem,
36      area_constraint sys ∧
37      parallel_constraint sys ∧
38      BD_ratio_constraint sys ∧
39      solution_constraint sys →
40      solution = sys.sol_area_CDE
```

Figure E-10: Case study 1 to illustrate the nuanced inconsistencies in MATH-ASCC-Eval-150

The constraint in the red box does not require that all x satisfying the condition are in the possible_x set, but only mandates that elements in possible_x must satisfy the condition. This is inconsistent with the semantics of the original problem. ASCC-3-MV successfully captured this semantic discrepancy.

**Problem:**
If $x^2 - x - 1 = 0$, what is the value of $x^3 - 2x + 1$?

```
 1  -->>declare_auxiliary_types
 2  -- No auxiliary types needed for this problem
 3
 4  -->>declare_mathematical_system
 5  structure MySystem where
 6    possible_x : Set ℝ
 7    sol_values : Set ℝ
 8
 9  -->>declare_auxiliary_functions_or_predicates
10  def cubic_expression (x : ℝ) : ℝ := x^3 - 2*x + 1
11
12  -->>declare_constraints
13  def quadratic_equation_constraint (sys : MySystem) : Prop :=
14    ∀ x ∈ sys.possible_x, x^2 - x - 1 = 0
15
16  def solution_constraint (sys : MySystem) : Prop :=
17    sys.sol_values =
18    {y | ∃ x ∈ sys.possible_x, y = cubic_expression x}
19
20  -->>declare_the_problem_statement
21  def problem_statement (solution : Set ℝ) : Prop :=
22    ∀ sys : MySystem,
23      quadratic_equation_constraint sys ∧
24      solution_constraint sys →
25      solution = sys.sol_values
```

Figure E-11: Case study 2 to illustrate the nuanced inconsistencies in MATH-ASCC-Eval-150

## APPENDIX F  CASE STUDIES FOR ITERATIVE RECTIFICATION ON MATH-50

As illustrated in Figure F-13, the top half of the diagram demonstrates the output of our method during the iterative rectification process. The left portion shows the generated result prior to rectification, which is inconsistent with the original problem. This inconsistency arises because the triangle constraint does not prohibit the points from being collinear. As illustrated in the right portion of the figure, the ASCC successfully detected and corrected this discrepancy (among others) in subsequent iterations, whereas the baseline method made the same error without detecting it.The LLM response generated by DeepSeek-V3 is as F-12

---

**ASCC Check Result**

```
{"discrepancies": [
    "The orthocenter is incorrectly defined as the vector sum of the triangle's vertices,
violating the correct geometric definition of the orthocenter as the intersection point of the
altitudes.", "The circumradius R is included in the MySystem structure but is not used in any
constraints or the solution, despite being required in the original problem.",
    "The solution sol_sum_squares is calculated purely geometrically without being expressed
in terms of a, b, c, and R as required by the original problem. Additionally, the orthocenter
calculation is incorrect.",
    "The orthocenter H is not uniquely determined by the given constraints, violating the
Exhaustion Principle of Predicate Satisfaction. The placeholder definition `sys.H = vec_add A
(vec_add B C)` does not correctly compute the orthocenter."],
"recommendations": [
    "Replace the placeholder definition of the orthocenter with a correct geometric
definition. For example, use the formula for the orthocenter in terms of the vertices and the
circumradius, or implement a function that computes the intersection point of the altitudes.
The corrected definition should comply with the Mandatory Lean4 Template.",
    "Add a constraint that relates R to the triangle ABC, such as defining it as the
circumradius of the triangle. For example, add a new constraint `circumradius_constraint` that
computes R from the triangle's side lengths and angles or uses existing Mathlib functions to
enforce this relationship.",
    "Modify the solution_constraint to express sol_sum_squares in terms of a, b, c, R using
the known formula (9R² - (a² + b² + c²)/2). Also, implement a correct orthocenter
calculation.",
    "Replace the placeholder definition with a correct computation of the orthocenter. For
example, the orthocenter can be computed as the intersection point of the altitudes of the
triangle. Here is a corrected definition:
```lean4
def orthocenter_constraint (sys:MySystem) : Prop :=
  let ⟨A, B, C⟩ := sys.triangle
  let altitude_A := ...  -- Compute the altitude from A to BC
  let altitude_B := ...  -- Compute the altitude from B to AC
  sys.H = intersection_point altitude_A altitude_B  -- Assuming a function to compute
intersection points
```
Note: The exact implementation of the altitude and intersection point computations would
depend on the available geometric functions in the Mathlib library."],
"consistency_level": "level_3"}
```

Figure F-12: LLM response during ASCC check

**Problem：**

Let $G$ and $H$ denote the centroid and orthocenter of triangle $ABC$, respectively. Let $F$ be the midpoint of $\overline{GH}$. Express $AF^2 + BF^2 + CF^2$ in terms of the side lengths a, b, c and circumradius R of triangle $ABC$.

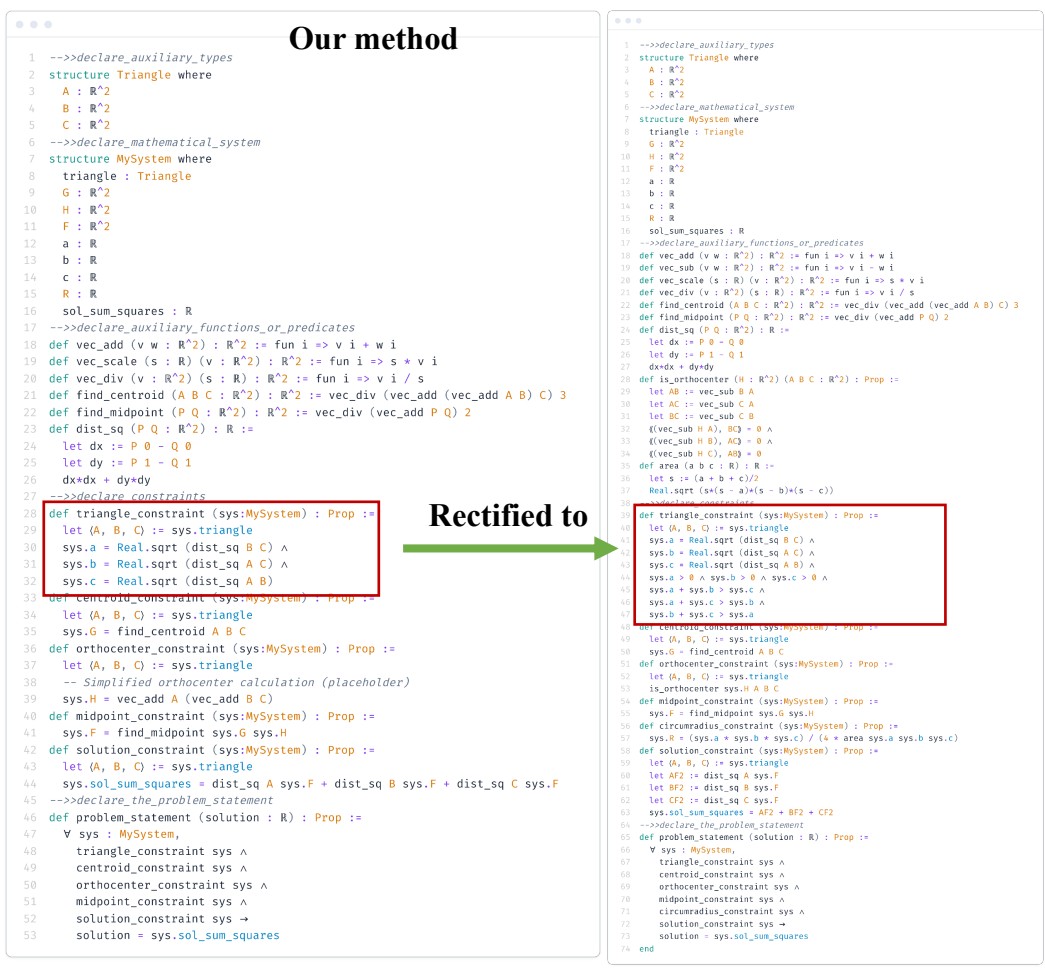

Figure F-13: Case study 3 to illustrate the rectification process compared to baseline

## APPENDIX G  ADDITIONAL EXPERIMENTAL RESULTS

To further eliminate potential biases from the dataset and the base model used for data generation, we performed manual annotation on 50 samples randomly selected from the outputs of KIMI-K2 on the miniF2F dataset. The sample set comprises 25 cases labeled as correct by ASCC-3-MV and 25 cases labeled as incorrect.

The annotation results show that among the samples deemed correct by ASCC, all passed human verification. In contrast, among those marked as incorrect by ASCC, 9 were judged as correct by human annotators. This finding aligns with the conclusions drawn from ASCC-Eval-150: the ASCC method exhibits a low false positive rate and a relatively high but acceptable false negative rate. These results, presented in Table 7, indicate that ASCC serves as a stringent evaluation criterion, which is advantageous for our objective of identifying errors for feedback and correction. Moreover, they further affirm that the accuracy metric derived from ASCC-3-MV provides a reliable and credible assessment.

Table 7: Evaluation of ASCC-3-MV on results generated by KIMI-K2

| Items | ASCC-positive | ASCC-negative |
|---|---|---|
| Human-positive | 25 | 9 |
| Human-negative | 0 | 16 |

We also conducted a comparative experiment using Bidirectional Equivalence(BEq) as the evaluation metric. The evaluation was limited to the LoC-Decomp method with KIMI-K2 as the base model, along with the best-performing baseline in the LLM-as-a-judge setup, Goedel-Autoformalizer-V2-32B. The results are summarized below in Table 8.

Table 8: Evaluation using BEq on mini-F2F

| Items | LoC-Decomp | Goedel-Autoformalizer-V2-32B |
|---|---|---|
| mini-F2F | 62.82 | 54.91 |

The BEq-based evaluation reveals notable differences in absolute pass rates compared to the LLM-as-a-judge approach, which is expected given the probabilistic nature of LLM judgments and the current limitations of automated theorem proving. Nevertheless, in terms of relative performance, the BEq results align with those from LLM-as-a-judge: our method continues to show a clear advantage over fine-tuned specialized models.

Detailed experimental setup: To perform BEq evaluation, it is necessary to formulate a bidirectional implication theorem and supply its proof, as illustrated below:

Code 2: Lean 4 theorem to prove the bidirectional equivalence

```
theorem bidirectional_equivalence :
    generated_proposition ↔ miniF2F_original_proposition := by
```

Here, `miniF2F_original_proposition` denotes the original proposition from the miniF2F dataset, while `generated_proposition` refers to the model-generated proposition. Since both the miniF2F dataset and the Lean4 code produced by Goedel-autoformalizer-V2 are structured as `theorem` statements awaiting proof, we first transformed them into predicate forms using the `def` keyword. This conversion was carried out using the DeepSeek-V3.1 model.

After constructing the `bidirectional_equivalence` theorem, we employed DeepSeek-V3.1 as an automated theorem prover. The prover was allowed up to 10 iterative corrections in case of compilation errors. Only proofs that passed the compiler without errors were considered valid.

We also report the self-check pass rates—where "self-check" refers to a single ASCC evaluation performed by the current model itself during the iterative process, as opposed to DeepSeek-V3 model usd in ASCC-3 metric—through one, two, and three rounds of iterative refinement on the MATH-500

and miniF2F datasets. The results show that the self-check pass rate increases with more iterations for both DeepSeek-V3 and KIMI-K2. In contrast, Qwen3-235B exhibits a significant deviation from our standard ASCC-3-MV metric. We argue this is because smaller-scale models struggle to make judgments on semantic consistency that align with human standards. See Figure G-14 for details.

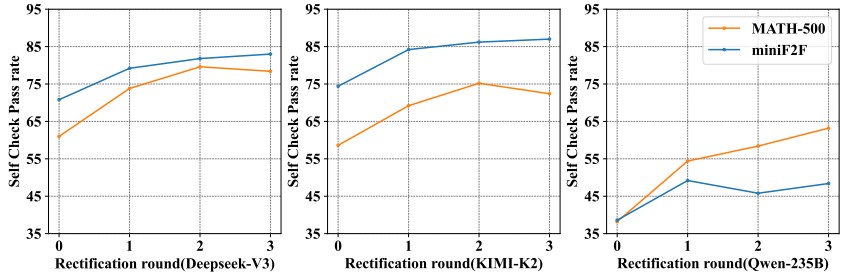

Figure G-14: Self-check pass rate during iteration

## APPENDIX H  DETAILS ABOUT OUR BASELINES

For both the Formal-Baseline and Formal-Iterative-Baseline, we adopted exactly the same example set and few-shot mechanism as our approach, except that for the baseline method, the formal Lean 4 code is transformed into a theorem with sorry like previous works(Azerbayev et al., 2023a; Li et al., 2024a; Wu et al., 2022). For the Formal-Iterative-Baseline, the iteration settings are exactly the same with our method, except it adopted a basic semantic checker. The SFT-Baseline used the same configuration as the Formal-Baseline, with the exception that the base model was replaced by DeepSeek-Prover-V2.

## APPENDIX I  PROMPTS

### I.1  PROMPT DIFFERENCES ACROSS MODELS

We observed that the Qwen3-235B model tends to generate overly lengthy responses (resembling a thinking mode despite being used as an instruct model). To address this, we introduced instructions such as *Your analysis must be concise but accurate.* and *Note: Avoid tediously long analysis.* to guide the model toward producing more concise outputs. The inclusion of these instructions successfully reduced verbosity in the model's responses.

We further evaluated the effect of these instructions on DeepSeek-V3 and KIMI-K2. The results indicated that DeepSeek-V3 generated more concise content when the instructions were applied, which led to a performance drop on the ASCC evaluation using the MATH-150 dataset. In contrast, KIMI-K2 was largely unaffected by the instructions, demonstrating greater robustness to minor prompt variations.

We argue that this discrepancy is reasonable, as differences in training strategies and data can lead to varied output styles across models. Although some minor differences on prompt is acceptable in practice, these findings highlight a limitation of our approach: it is susceptibility to specific prompt designs.

### I.2  THE PROMPT USED FOR FORMALIZATION

```
# Please use Lean 4 code to convert the textual description of the
    problem into a formal representation.
## Instruction
The following is a middle school mathematics problem. Please use Lean 4
    code to convert the textual description of the problem into a formal
    representation.
```

```
Any mathematical problem can be viewed as the study of the properties of
    a specific mathematical system. A mathematical system consists of
    several mathematical objects, along with a set of constraints that
    limit the properties or relationships of these objects, thereby
    defining the mathematical system under investigation. The
    requirement of the mathematical problem is to study a particular
    property of this mathematical system. Therefore, a mathematical
    problem can be formally expressed by declaring a mathematical system
    composed of several mathematical objects and a series of
    constraints, and for any concrete instance of the system, the
    conjunction of all these constraints imply a statement about the
    solution.

## Attentions
Note: You do not need to solve the problem, your task is solely to
    translate the problem's description into formal Lean 4 code.
Note: For the naming of any symbols, if the stem explicitly provides a
    name, it should be consistent with the stem; otherwise, provide a
    reasonable name.
Note: The sole purpose of naming is to improve code readability. Names
    cannot be relied upon to convey semantic meaning or mathematical
    structure all such information must be expressed through the code's
    logic.
Note: Please pay attention to distinguishing the syntax differences
    between Lean 3 and Lean 4.

## Problem
The problem to be processed is as follows:
{original_problem}

## Response Template
Here, you should formalize the above problem in Lean4, strictly
    following the provided Lean4 template as shown below.
'''Lean4
import Mathlib

-->>declare_enviroment
open Real InnerProductGeometry Matrix Topology Filter ENNReal Polynomial
    Classical Complex

notation "ℝ^"n => EuclideanSpace ℝ (Fin n)
notation "𝕄 [" m "," n "]" => Matrix (Fin m) (Fin n) ℝ
notation "⟨⟨" x ", " y "⟩⟩" => @inner ℝ _ _ x y

variable {V : Type} [NormedAddCommGroup V] [InnerProductSpace ℝ V]

/-
The foundational environment has been established, and you can not
    modify it. For the sake of automated control, declaring additional
    notations or opening more namespaces is not allowed.
-/

noncomputable section
-->>declare_auxiliary_types
/-
Some mathematical objects have complex types and need to be declared
    using the 'structure' or 'inductive' keywords. At this stage, you
    can declare the types of certain complex mathematical objects for
    ease of later use.
-/

-->>declare_mathematical_system
/-
```

```
In Lean 4, declaring a mathematical system involves defining a structure
    (must be named as 'MySystem' for automatic code check) composed of
    mathematical objects that may represent either structural spaces
    (possible value domains) or concrete instances (specific values).
    The mathematical system consists of several mathematical objects,
    along with a set of constraints that limit their properties or
    relationships. Follow the below principles when constructing this
    system:

1. Complete System Declaration Principle:
   - For readability, the mathematical system structure must contain all
       objects mentioned or implied in the problem statement, which
       enables immediate visual mapping between problem text and formal
       structure, and help the readers to understand.

2. Type Specification Principle:
   - Declare only the types of mathematical objects at this stage.
   - Detailed definitions and property relationships will be established
       later through constraint predicates.

3. Solution Object Principle:
   - There must exists a special mathematical object that named with the
       prefix 'sol_',denotes the target property/solution the problem
       seeks
   - Analyze the problem to determine:
     * What constitutes a valid solution
     * Whether it represents a concrete value or solution space
     * The appropriate type representation (set/type for spaces, direct
        type for unique solutions)

4. Clarify Point-Space Principle
   - A mathematical object should be formalized as a concrete instance
       (e.g., a : Nat) if and only if it is uniquely determined within
       the problem's constraints. Otherwise, it must be represented as a
       set or type (e.g., Set Nat) to preserve its possible value space.

5. Exhaustion Principle of Predicate Satisfaction
   - If it is desired for set A to contain all elements that satisfy
       predicate P, it must be ensured that every element in set A
       satisfies predicate P, and that there does not exist an element e
       such that e satisfies predicate P but e does not belong to set A.
-/

-->>declare_auxiliary_functions_or_predicates
/-
The definitions, properties, or relationships of declared objects need
    to be specified later by declaring predicate constraints. However,
    some predicates may be overly complex, so you can define auxiliary
    functions or predicates at this stage to simplify the subsequent
    predicate constraints. Predicates defined here cannot serve as
    antecedents in problem_statement implications. They require
    encapsulation as constraint predicates in the next step before usage.

For the purpose of automated control, parameters of type MySystem cannot
    be used in this step.
-/

-->>declare_constraints
/-
In this step, you need to express the definitions, properties, or
    relationships of mathematical objects in the system by declaring
    predicates. The names of these predicates must end with
    '_constraint' for better readability.
```

```
All constraints from the original problem must be reflected in this
    step, and all constraint predicates declared here must appear in the
    problem statement as antecedents in implication expressions.

While maintaining strict logical rigor, prioritize code readability and
    semantic clarity by avoiding the conflation of multiple constraints
    within a single restrictive predicate. For the sake of code
    readability and semantic clarity, a reasonable amount of code
    redundancy is acceptable. Avoid sacrificing readability in pursuit
    of excessive brevity.

Additionally, the constraint describing the property of the `sol_`
    object should be named `solution_constraint`, which is used to
    constrain the `sol_` object so that it satisfies the requirements of
    the original problem and becomes the solution to it.
-/

-->>declare_the_problem_statement
/-
In this step, you need to declare a special predicate called
    `problem_statement`, which takes the `solution` variable as a free
    input variable. This predicate applies all constraint predicates to
    the mathematical system, thereby completing the declarative
    expression of the original problem. The format of the
    `problem_statement` predicate is strictly defined: for any
    mathematical system, the following implication holds, where the
    antecedent is the logical AND of all constraint predicates, and the
    consequence is `solution = sol_object`.

The semantics of `problem_statement` are as follows: the proposition is
    true if and only if the value of `solution` is the solution to the
    original problem.
-/

end
```
```

## Examples
```

I.3 THE PROMPT USED FOR BACK TRANSLATION (PER CONSTRIANTS)

```
## Requirement : Translate Lean4 to natural language
I am attempting to formalize a certain mathematical problem. To achieve
    this formalization, I model the mathematical problem as a
    mathematical system consisting of several mathematical objects and a
    series of constraints on the system. The following Lean4 code
    represents one of the complete formalized constraints for the
    mathematical problem. Additionally, MySystem describes the
    mathematical system, while the rest of the code serves as auxiliary.
    Please analyze the semantics of this constraint and express it in
    natural language.Pay attention the use of quantifiers and the domain
    of variables.

Any mathematical problem can be viewed as the study of the properties of
    a certain mathematical system. A mathematical system consists of
    several mathematical objects, along with a series of constraints
    that limit the properties or relationships of these mathematical
    objects, thereby defining the mathematical system under study. The
    requirement of a mathematical problem is to investigate a certain
    property of this mathematical system. Therefore, a mathematical
    problem can be formally expressed as declaring a mathematical system
    composed of several mathematical objects and a series of constraints.
```

```
Note: There is some distinctions between certain Lean4 data structures,
    for example, between List and Set, where a List may contain
    duplicate elements, while a Set enforces uniqueness, reflect the
    feature of the data structure used.
Note: Your task is solely describe the semantic meaning of the
    constraint's Lean4 code, do not make any calculations or
    derivations. Focus exclusively on providing a precise mathematical
    characterization of the system formalized in this code. Do not make
    any assumptions about the real-world problem it might represent.
Note: You must carefully translate the logical structure of Lean4 code,
    especially when it involves universal quantifiers or existential
    quantifiers.
Note: The naming of objects is merely an identifier and should not be
    used as a reference for semantic meaning.
Note: Describing Lean4 code in natural language may introduce ambiguity,
    so try your best to avoid ambiguity.
Note: If any auxiliary functions are used in the Lean4 code of a
    constraint, describe their semantic meaning integrated naturally
    with the constraint, rather than referring to them by name.

Note: In your analysis and summary, you must reflect the below two
    principles(Clarify Point-Space Principle and Exhaustion Principle of
    Predicate Satisfaction).
## Formalization Principle
Clarify Point-Space Principle
    - A mathematical object should be formalized as a concrete instance
        (e.g., a : Nat) if and only if it is uniquely determined within
        the problem's constraints. Otherwise, it must be represented as a
        set or type (e.g., Set Nat) to preserve its possible value space.

Exhaustion Principle of Predicate Satisfaction
    - If it is desired for set A to contain all elements that satisfy
        predicate P, it must be ensured that every element in set A
        satisfies predicate P, and that there does not exist an element e
        such that e satisfies predicate P but e does not belong to set A.

@magic_method_or_not

## Your response should strictly follow the template below.
### **Analysis of `<constraint name>`**
<Your analysis of the constraint,must reflect the above two principles>

### **Concise Summary of `<constraint name>`**
```markdown
<Here is a clear and natural language summary of the constraint's
    meaning that incorporates the semantics of any auxiliary functions.
    Emphasize adherence to the above principles, especially in cases
    where they may have been overlooked.>
```

## Lean4 code you need to translate
```Lean4
<Lean4_code>
```
```

## I.4 THE PROMPT USED FOR CONSISTENCY LEVEL DETERMINATION

```
# Requirement: The Lean4 code is a formalization of
    problem_original(intendedly transformed into a verification task by
    introducing `{sol_obj}`,since there is no concept such as `solve` in
```

```
    Lean4), but it may have some issues in the formalization procedure.
    I have observed some semantic discrepancies between the Lean4 code
    and the problem_original but I can not be sure wether these
    discrepencies is severe enough to break the semantic consistency.
    Please analyse every discrepancy listed and determine the
    consistency level of the Lean4 code, by the final criterion and
    definitions of consistency level. Your analyse must be concise but
    accurate.

## **Consistency Criteria **
The Lean 4 code is deemed **consistent** with the original problem if
    and only if it **does not violate**  any of the following
    **Violation Criteria** .
### ** Violation Criteria **
*(If any of these are true, the code is **inconsistent**.)*
**Violation Criterion 1: Object Omission**
Not all mathematical objects in `problem_original` are reflected in the
    Lean4 code or their types mismatch(except for `{sol_obj}`, the type
    of `{sol_obj}` can be different but must be equivalent). As long as
    the objects are correctly reflected, some redundancy in the Lean4
    code are acceptable.

**Violation Criterion 2: Semantic Alteration**
The definitions, properties, or relationships of any mathematical object
    from `problem_original` are not exactly preserved in the Lean4 code.
    *(Note: Some redundancy is permitted as long as the core semantics
    are constrained correctly.)*

**Violation Criterion 3: Constraint Incompleteness**
The constraints expressed in problem_formal fail to comprehensively
    represent all explicit and implicit constraints present in
    problem_original. (Note: Full completeness is essential, both
    explicitly stated and implicitly inferred constraints must be
    accurately formalized.)

**Violation Criterion 4: Over-Simplification**
The Lean4 code conducts concrete computation or derivation that
    simplifies the original problem, leading to a semantic inconsistency.

### **Exceptions Criteria**
**Exception Criterion 1: Formatting Flexibility**
There is no need to strictly adhere to the original problem's formatting
    requirements for the solution, as Lean4 does not support extensive
    formatting options (such as requiring decimal numbers, etc.).

### **Final Criterion **
Your determination should focus on the discrepancies listed, other
    aspects are irrelevant.
The Lean 4 code is **consistent** with the original problem **if and
    only if**: All the discrepancies listed are acceptable under the
    above criteria.
As long as the core properties and relationships of the essential
    objects are preserved, some redundancy or unnecessary complexity in
    the Lean4 code is acceptable.

## Formalization Principle for reference:
Clarify Point-Space Principle
   - A mathematical object should be formalized as a concrete instance
       (e.g., a : Nat) if and only if it can be uniquely determined
       within the problem's constraints. Otherwise, it must be
       represented as a set or type (e.g., Set Nat) to preserve its
       possible value space.

Exhaustion Principle of Predicate Satisfaction
```

```
    - If it is desired for set A to contain all elements that satisfy
      predicate P, it must be ensured that every element in set A
      satisfies predicate P, and that there does not exist an element e
      such that e satisfies predicate P but e does not belong to set A.

## Mandatory Lean4 Template:
```Lean4
-->>declare_enviroment
<Some enviroment>
-->>declare_auxiliary_types
<some auxiliary types>
-->>declare_mathematical_system
-->>declare_mathematical_system
structure MySystem where
  <objects>
-->>declare_auxiliary_functions_or_predicates
<some auxiliary definitions, without any parameter be of type MySystem>
-->>declare_constraints
def <name>_constraint (sys:MySystem) : Prop :=
  <expr_body>
def <name>_constraint (sys:MySystem) : Prop :=
  <expr_body>
def solution_constraint (sys:MySystem) : Prop :=
  <expr_body>
-->>declare_the_problem_statement
def problem_statement (solution : <type>) : Prop :=
  ∀ sys:MySystem,
    <name>_constraint sys ∧
    <name>_constraint sys ∧
    solution_constraint sys →
    <expression of solution and sol_object>
end
```

## problem_original:
@original_problem

## Lean4 Code To Judge:
```Lean4
@Lean4_code
```

## Potential Sementic Discrepancies:
@potential_sementic_discrepancies

## Consistency level:
There are three levels of consistency you can choose.
level_1: Fully consistent by the final criterion.
level_2: Consistent without loss of generality: The Lean4 code
    formalizes the problem by analyzing representative cases. In each of
    these cases, the final criterion are fully satisfied, and the
    general case follows through straightforward deduction.Or the
    formalization rely on equivalent conversions, such as transforming
    canonical equations into general form. In such cases, the
    formulation can be regarded as consistent without loss of generality.
level_3: Inconsistent, any criterion breaked can lead to this.

Note: Your reasons and recommendations should avoid including any
    specific calculations or derivations, as this may lead to
    inconsistency.

Note: Avoid tediously long analysis.
## Your response should be like:
### Analyse of all the listed discrepancies:
```

```
<Analyse the discrepancies by examining the Lean4 code and original
    problem to determine wether it is severe engough to impair the
    semantic consistency. Make sure every discrepancy listed is analysed
    in detail.>

### Analyse Of each consistency level:
<According to the definitions of consistency level and your above
    analyse about the discrepancies listed, determine which consistency
    level is appropriate.Your determination should focus on the
    discrepancies listed, other aspects are irrelevant.>

### Consistency Level Determination
```json
{
"consistency_level" : "<level_1 or level_2 or level_3>",
"discrepancies" : ["<According to your previous analysis, list all the
    semantic discrepancies here. If consistency_level is `level_1`,
    leave this filed blank.Please provide some detailed descriptions of
    the discrepancies, ensuring it is concrete rather than high-level.
    Exclude discrepancies related to unnecessary abstractions,
    complexity, or redundancy, as that is not really matter.>"],
"recommendations" : ["<Some specific recommendations to rectify the
    Lean4 code instead of high-level recommendations such as rectify
    which declaration into what. Your recomentation must align with the
    Formalization Basement and Mandatory Lean4 Template. The
    recommendations must directly correspond to the discrepancies. If
    consistency_level is `level_1`, leave this filed blank.>"]
}
```
```

## I.5 THE PROMPT USED FOR RECTIFICATION

```
# Requirement: There appears to be a discrepancy between your Lean4
    formalization and the original problem, due to the following
    reasons, and some recommendations to rectify the Lean4 code are
    provided. Please rectify the Lean4 code accordingly. Print the
    rectified Lean4 code directly according to the error information. Do
    not make further thinking.
Note: All suggestions are indicative, not mandatory. Rectify the Lean4
    code based on your understanding.
Note: Your rectification must meet all the formalization principles
    mentioned above.
Note: Your formalization must adopt this treatment: the solution to the
    original problem is taken as a free variable in the proposition
    **problem_statement**, such that **problem_statement** holds true if
    and only if the **solution** is indeed a valid solution to the
    original problem.

The determination of consistency follows the criteria below.
## **Consistency Criteria **
The Lean 4 code is deemed **consistent** with the original problem if
    and only if it **does not violate**  any of the following
    **Violation Criteria** .
### ** Violation Criteria **
*(If any of these are true, the code is **inconsistent**.)*
**Violation Criterion 1: Object Omission**
Not all mathematical objects in `problem_original` are reflected in the
    Lean4 code or their types mismatch(except for `{sol_obj}`, the type
    of `{sol_obj}` can be different but must be equivalent). As long as
    the objects are correctly reflected, some redundancy in the Lean4
    code are acceptable.

**Violation Criterion 2: Semantic Alteration**
```

```
The definitions, properties, or relationships of any mathematical object
    from 'problem_original' are not exactly preserved in the Lean4 code.
    *(Note: Some redundancy is permitted as long as the core semantics
    are constrained correctly.)*

**Violation Criterion 3: Constraint Incompleteness**
The constraints expressed in problem_formal fail to comprehensively
    represent all explicit and implicit constraints present in
    problem_original. (Note: Full completeness is essential both
    explicitly stated and implicitly inferred constraints must be
    accurately formalized.)

**Violation Criterion 4: Over-Simplification**
The Lean4 code conducts concrete computation or derivation that
    simplifies the original problem, leading to a semantic inconsistency.

### **Exceptions Criteria**
**Exception Criterion 1: Formatting Flexibility**
There is no need to strictly adhere to the original problem's formatting
    requirements for the solution, as Lean4 does not support extensive
    formatting options (such as requiring decimal numbers, etc.).

### **Final Criterion **
Your determination should focus on the discrepancies listed, other
    aspects are irrelevant.
The Lean 4 code is **consistent** with the original problem **if and
    only if**: All the discrepancies listed are acceptable under the
    above criteria.
As long as the core properties and relationships of the essential
    objects are preserved, some redundancy or unnecessary complexity in
    the Lean4 code is acceptable.

Reflect the Clarify Point-Space Principle and the Exhaustion Principle
    of Predicate Satisfaction in your analysis. For some cases,
    violation of these principles may lead to severe semantic
    inconsistencies.

Clarify Point-Space Principle
    - A mathematical object should be formalized as a concrete instance
        (e.g., a : Nat) if and only if can be uniquely determined within
        the problem's constraints. Otherwise, it must be represented as a
        set or type (e.g., Set Nat) to preserve its possible value space.

Exhaustion Principle of Predicate Satisfaction
    - If it is desired for set A to contain all elements that satisfy
        predicate P, it must be ensured that every element in set A
        satisfies predicate P, and that there does not exist an element e
        such that e satisfies predicate P but e does not belong to set A.

Final criterion:
The Lean 4 code is consistent with the original problem if and only if,
    for any concrete instance of MySystem satisfying the constraints,
    the three basic criteria are satisfied. As long as the properties or
    relationships of those essential objects are not compromised, some
    redundancy or unnecessary complexity in the Lean4 code is acceptable
    as long as the above three criteria are fully satisfied.

## Reasons:
@reasons

## Recomendations:
@recommendations

@previous_recommendations
```

```
## Your response should strictly follow the following template:

'''Lean4
import Mathlib

-->>declare_enviroment
open Real InnerProductGeometry Matrix Topology Filter ENNReal Polynomial
    Classical Complex

notation "ℝ^"n => EuclideanSpace ℝ (Fin n)
notation "𝕄 [" m "," n "]" => Matrix (Fin m) (Fin n) ℝ
notation "⟨⟨" x ", " y "⟩⟩" => @inner ℝ _ _ x y

variable {V : Type} [NormedAddCommGroup V] [InnerProductSpace ℝ V]

/-
The foundational environment has been established, and you can not
    modify it. For the sake of automated control, declaring additional
    notations or opening more namespaces is not allowed.
The purpose of this formalization project is for educational purposes,
    so we strive to minimize reliance on implementations from the
    Mathlib library. Apart from 'import Mathlib', no additional imports
    should be included.
-/

noncomputable section
-->>declare_auxiliary_types
<some auxiliary types>
-->>declare_mathematical_system
structure MySystem where
  <objects>
-->>declare_auxiliary_functions_or_predicates
<some auxiliary definitions, without any parameter be of type MySystem>
-->>declare_constraints
def <name>_constraint (sys:MySystem) : Prop :=
  <expr_body>
def <name>_constraint (sys:MySystem) : Prop :=
  <expr_body>
def solution_constraint (sys:MySystem) : Prop :=
  <expr_body>
-->>declare_the_problem_statement
def problem_statement (solution : <type>) : Prop :=
  ∀ sys:MySystem,
    <name>_constraint sys ∧
    <name>_constraint sys ∧
    solution_constraint sys →
    <expression of solution and sol_object>
end
'''
```

## I.6 THE PROMPT USED FOR SC-BASELINE

```
You will receive a natural language math problem statement, along with
    its formal statement
in LEAN 4 and, in some cases, a description of mathematical terms.
    Please evaluate whether
the formal LEAN statement appropriately translates the natural language
    statement based on
the following criteria. They are considered different if any of the
    criteria are not satisfied.
1. Key Elements: The fundamental mathematical components, including
    variables,
```

```
constants, operations, domain, and codomain are correctly represented in
    LEAN code.
2. Mathematical Accuracy: The mathematical relationships and expressions
    should be
interpreted consistently during translation.
3. Structural Fidelity: The translation aligns closely with the original
    problem, maintaining
its structure and purpose.
4. Comprehensiveness: All conditions, constraints, and objectives stated
    in the natural
language statement are mathematically included in the LEAN translation.
When doing evaluation, break down each problem statement into
    components, match the
components, and evaluate their equivalence.
Think step-by-step and explain all of your reasonings.

Natural language math problem statement:
{original_problem}

Formal statement in LEAN 4
{Lean4_code}
Your answer should be like:
<some analyase here>

Judgement and recommendation:
```json
{{
    "consistent_or_not" : "<yes or no>",
    "reasons" : "<if inconsistent, leave your reasons here>",
    "recommendations" : "<if inconsistent, leave your recommendations
        here>"
}}
```
```

## I.7 THE PROMPT USED FOR SC-BASELINE-BT

```
You will receive a natural language math problem statement, along with
    its formal statement
in LEAN 4 and, in some cases, a description of mathematical terms.
    Please evaluate whether
the formal LEAN statement appropriately translates the natural language
    statement based on
the following criteria. They are considered different if any of the
    criteria are not satisfied.
1. Key Elements: The fundamental mathematical components, including
    variables,
constants, operations, domain, and codomain are correctly represented in
    LEAN code.
2. Mathematical Accuracy: The mathematical relationships and expressions
    should be
interpreted consistently during translation.
3. Structural Fidelity: The translation aligns closely with the original
    problem, maintaining
its structure and purpose.
4. Comprehensiveness: All conditions, constraints, and objectives stated
    in the natural
language statement are mathematically included in the LEAN translation.
When doing evaluation, break down each problem statement into
    components, match the
components, and evaluate their equivalence. Think step-by-step and
    explain all of your
reasonings. Your answer should be in the following format:
Thought: [Your Answer]
```

```
Judgement: [Your Answer, one of Appropriate, Inappropriate]

Natural language math problem statement:
{original_problem}

Formal statement in LEAN 4
{Lean4_code}

Natural language description of LEAN 4:
{back_translation}
```

## I.8   THE PROMPT USED FOR FORMAL-BASELINE

```
Please translate the following mathematical problem into Lean4 by
    completing the template provided. Do not try to solve the problem,
    your task is formalization. Do not make any derivation or comutation
    as that would be inconsistent with the original problem. Enclose
    your Lean4 code in a ```Lean4``` section.

Problem:
{problem}

Lean4 tamplate to complete:
```Lean4
import Mathlib

open Real InnerProductGeometry Matrix Topology Filter ENNReal Polynomial
    Classical Complex

notation "ℝˆ"n => EuclideanSpace ℝ (Fin n)
notation "𝕄 [" m "," n "]" => Matrix (Fin m) (Fin n) ℝ
notation "⟨⟨" x ", " y "⟩⟩" => @inner ℝ _ _ x y

variable {V : Type} [NormedAddCommGroup V] [InnerProductSpace ℝ V]

noncomputable section
theorem <name> <parameters> (solution:<type of solution>) : <conclusion>
    := by sorry

end

Examples:

```
```

## I.9   THE PROMPT USED FOR FORMAL-ITERATIVE-BASELINE(SEMANTIC RECTIFICATION)

```
# Your Lean4 code is not consistent with the original problem for the
    following reasons, and try to rectify your Lean4 code according to
    the reasons and recommendations.

## Reasons:
{reasons}

## Recommendations:
{recommendations}

Lean4 tamplate to complete:
```

```Lean4
import Mathlib

open Real InnerProductGeometry Matrix Topology Filter ENNReal Polynomial
    Classical Complex

notation "ℝ^"n => EuclideanSpace ℝ (Fin n)
notation "𝕄 [" m "," n "]" => Matrix (Fin m) (Fin n) ℝ
notation "⟨⟨" x ", " y "⟩⟩" => @inner ℝ _ _ x y

noncomputable section
theorem <name> <parameters> (solution:<type of solution>) : <conclusion>
    := by sorry

end
```

## I.10   THE PROMPT USED FOR FORMAL-ITERATIVE-BASELINE(SYNTACTIC RECTIFICATION)

```
# Requirement: There appears to be some errors in your Lean4 code.
    Please rectify the Lean4 code accordingly.
Note: You only need to correct syntax errors, do not change any
    declaration's name, parameters, type, or semantic, otherwise the
    Lean4 code will be inconsistent with the original problem.
Note:Try to implement the same logic using simpler syntax by yourself,
    rather than relying on Lean4 or Mathlib's built-in special features,
    to reduce the possibility of syntax errors or compilation errors.

## Errors:
@err_msg_str

## Your response must follow the following template:

```Lean4
import Mathlib

open Real InnerProductGeometry Matrix Topology Filter ENNReal Polynomial
    Classical Complex

notation "ℝ^"n => EuclideanSpace ℝ (Fin n)
notation "𝕄 [" m "," n "]" => Matrix (Fin m) (Fin n) ℝ
notation "⟨⟨" x ", " y "⟩⟩" => @inner ℝ _ _ x y

noncomputable section
theorem <name> <parameters> (solution:<type of solution>) : <conclusion>
    := by sorry

end
```
```

