# OpenReview forum: "LoC-Decomp: LLM Autoformalization via Logical Concept Decomposition and Iterative Feedback Correction"
_ICLR.cc/2026/Conference — ICLR 2026 Poster_

### Official Review · Reviewer_2crV · 2025-10-30

**Soundness:** 3
**Presentation:** 3
**Contribution:** 3
**Rating:** 6
**Confidence:** 5

**Summary:**

This paper introduces **Loc-Decomp**, a framework that improves automated formalization—the process of converting natural language math statements into formal, machine-verifiable code. It combines a **structured Lean4 template**, a **semantic consistency checker**, and an **iterative feedback loop** that jointly refine both meaning and syntax. Unlike previous methods, Loc-Decomp detects subtle semantic mismatches and uses compiler errors to iteratively correct them. Experiments on MATH and miniF2F datasets show major improvements, achieving up to **90% formalization accuracy**, far surpassing previous state-of-the-art methods.

**Strengths:**

- A **divide–conquer–merge–based semantic self-checking mechanism** is proposed to detect subtle inconsistencies between the formalization and the original statement, which aligns well with the motivation behind **Retrieval-Augmented Generation**.
- For the first time, it combines semantic inconsistency feedback with compiler error information to perform iterative refinement.
- The method is **simple and easy to understand**.

**Weaknesses:**

- The experiments are too limited; please provide **comparative experiments on the ProofNet and Putnam datasets**.
- Please provide a **detailed ablation study** of **Logical Concept Decomposition** and **Iterative Feedback Correction**, explicitly isolating and analyzing their individual effects.

**Questions:**

Please Refer to Weaknesses

---

> ### Author Response · Authors · 2025-11-20
> **Response to Reviewer 2crV (1/1):**
>
> Dear Reviewer **2crV**,
>
> Thank you for your patience and constructive suggestions. In response to your review, we have made significant revisions to strengthen our work.
>
> **Response to weakness 1: limited experiments**
>
> We acknowledge the reviewer's concern regarding the limitations of our experiments. In response, we have supplemented our evaluation with additional experiments on the ProofNet and PutnamBench datasets, employing ASCC-3-MV as the evaluation metric.
>
> **Evaluation Results on Undergraduate-Level Benchmarks**
>
> Our method was further evaluated on two undergraduate-level datasets, PutnamBench and ProofNet, where it demonstrated strong performance despite a relative drop compared to high-school-level benchmarks. Specifically, it achieved ASCC-3-MV pass rates of up to 81.50% on PutnamBench and 70.57% on ProofNet. These results not only affirm the effectiveness of our approach on undergraduate-level mathematical problems but also empirically demonstrate its capacity to generalize to more advanced mathematical domains.
>
> -----
> **Without iterative rectification, ASCC-3-MV as metric**
>
> | Items           | DeepSeek-V3 |    | KIMI-K2 | |
> | :--- | :---: | :---: | :---: | :---: |
> |                 | **MV** | **SV** | **MV** | **SV** |
> | **PutnamBench** | 48.37  | 31.71  |  __53.66__ | 35.16  |
> | **ProofNet**    | 38.42  | 27.79  | __43.05__  | 29.16  |
>
> **With iterative rectification, ASCC-3-MV as metric**
>
> | Items | DeepSeek-V3 | | KIMI-K2 | |
> | :--- | :---: | :---: | :---: | :---: |
> | | **MV** | **SV** | **MV** | **SV** |
> | **PutnamBench** | __81.50__ | 58.33 | 72.36 | 48.78 |
> | **ProofNet** | 67.03 | 48.50 | __70.57__ | 52.59 |
> ----
>
> **Response to weakness 2: About ablation study**
>
> We appreciate the reviewer's suggestion regarding the ablation studies. We understand the importance of a clear presentation. To that end, we have enhanced the manuscript by providing a more detailed explanation of the experimental design in the relevant section. This expansion directly addresses the feedback and aims to offer a fuller context for the existing ablation studies presented in Tables 1-3.
>
> In the ablation study on Logical Concept Synthesis, the model was configured to use a pass@1 formalization process with a limit of three syntax correction rounds, while iterative semantic correction was disabled. The semantic consistency of the generated formalizations was then evaluated using the ASCC-3-MV metric. The corresponding results are presented below.
>
> **Results without iterative rectification, ASCC-3-MV as metric**
>
> | Items       | DeepSeek-V3 |         | KIMI-K2 |         | Qwen3-235B |         |
> |:----|:------:|:------:|:-----:|:----:|:-------:|:------:|
> |             | MV          | SV      | MV      | SV      | MV        | SV      |
> | **MATH-500**  | 63.20       | 44.80   | 60.20   | 43.40   | __65.40__ | 45.60   |
> | **miniF2F**   | 77.25       | 54.30   | __78.57__ | 60.29   | 76.90     | 60.29   |
>
>
> Iterative Feedback Correction extends this approach by adding iterative semantic correction, integrated with the syntactic correction per Section 3. We evaluated the semantic consistency of the results after three outer-loop iterations using the ASCC-3-MV metric; these results are shown below.
>
> **Results with iterative rectification, ASCC-3-MV as metric**
> | Items       | DeepSeek-V3 |         | KIMI-K2 |         | Qwen3-235B |         |
> |:---|:---:|:---:|:---:|:---:|:---:|:---:|
> |             | MV          | SV      | MV      | SV      | MV        | SV      |
> | **MATH-500**  | __84.00__   | 61.60   | 77.80   | 55.60   | 75.60     | 57.80   |
> | **miniF2F**   | 90.16       | 67.00   | __91.60__ | 72.90   | 80.25     | 61.07   |
>
>
> Additionally, we conducted an ablation study on Logical Concept Decomposition and Iterative Feedback Correction on the MATH-EVAL-50 dataset through manual inspection. In Table 3, the "Ours-no-iter" column represents the experimental results using only Logical Concept Decomposition, while the "Ours-iter" column shows the results after further incorporating Iterative Feedback Correction. The experimental results demonstrate that relying solely on Logical Concept Decomposition already outperforms the baseline method, and **the introduction of Iterative Feedback Correction further improves the pass rate of semantic consistency checks**, consistently observed in both ASCC-3-MV and manual evaluation results.
>
>
> | Items        | Ours-no-iter | Ours-iter | Baseline | Baseline-iter | DeepSeek-Prover-V2 |
> |--------------|:-------------:|:----------:|:---------:|:--------------:|:-------------------:|
> | pass rate    | 70.00        | __84.00__     | 52.00    | 54.00         | 66.00              |
>
> **Response to Questions**
>
> Please refer to Response to weakness.

---

> ### Author Response · Authors · 2025-11-25
> **Summary of responses:**
>
> **Weakness 1 (Narrow Scope of Evaluation):**
> We have supplemented our evaluation with experiments on undergraduate-level datasets including Proofnet and PutnamBench. The results show that our method still achieves strong performance on these datasets.
>
> **Weakness 2 (Ablation Study):**
> We have elaborated on the design of the ablation experiments in the experimental section of the manuscript, analyzing the contributions of Logical Concept Decomposition and Iterative Feedback Correction to the final results.
>
> **Questions:** The same as weaknesses.
>
> **Summary:**
> We think we have directly and comprehensively addressed all two weaknesses and questions raised by the reviewer.
>
> **Closing Remarks:**
> We greatly appreciate the reviewer’s constructive feedback and would welcome the opportunity to further discuss any remaining concerns or potential discrepancies.

---

> > ### Comment · Reviewer_2crV · 2025-11-26
> >
> > Thank you for providing the experiment result— I’ve accordingly improved my score.

---

> > > ### Author Response · Authors · 2025-11-26
> > >
> > > Thank you for your constructive suggestions and efforts towards the positive development of the academic community!

---

### Official Review · Reviewer_DZ3h · 2025-10-30

**Soundness:** 2
**Presentation:** 3
**Contribution:** 3
**Rating:** 4
**Confidence:** 3

**Summary:**

This paper introduces LoC-DeCOMP, a framework for automated formalization of mathematical statements using off-the-shelf LLMs. The approach integrates a structured decomposition template for Lean4 code generation, a semantic self-checking mechanism (ASCC) leveraging divide-conquer-merge back-translation, and an iterative feedback-driven refinement loop employing both semantic and syntactic (compiler) error signals. Experimental evaluation on the MATH-500 and miniF2F datasets demonstrates substantial improvements over baseline and prior state-of-the-art methods. Furthermore, experiments on the human-verified MATH-Level5-50 subset show a notable 30 percentage point improvement over the baseline.

**Strengths:**

1.	Clarity and Presentation: The paper is exceptionally well-written and easy to follow. The proposed method, LoC-DeCOMP, and its evaluation are described with great clarity, which significantly aids in understanding the contribution.
2.	Accessibility and Ease of Use: A key strength of this work is that it proposes a training-free workflow. By designing a system that effectively orchestrates off-the-shelf LLMs, the framework is highly accessible and can be readily implemented without the need for costly model fine-tuning.
3.	Demonstrated Effectiveness: The method shows significant empirical success. The 30 percentage point improvement over the baseline on the human-verified MATH-Level5-50 dataset is impressive and underscores the practical utility and effectiveness of the proposed workflow.

**Weaknesses:**

1.	Potential Loss of Expressive Capabilities: The framework's reliance on a predefined template for formalization, as mentioned in Appendix A.4, is a central concern. This template-based approach may inherently restrict the expressive power of the Lean4 language. The paper currently lacks a thorough analysis of this trade-off. Specifically, there is no justification or evidence to suggest that the template is sufficient for formalizing mathematics beyond the scope of high-school-level problems. If the template's expressiveness is severely limited, it could significantly impact the generalizability and overall contribution of the proposed method.
2.	Narrow Scope of Evaluation: The empirical evaluation is confined to the MATH and miniF2F datasets, both of which primarily consist of high-school-level mathematics. The paper would be much stronger with experiments on more advanced or diverse mathematical domains (e.g., undergraduate-level abstract algebra or analysis). This weakness is particularly concerning when considered alongside the potential loss of expressiveness (Weakness 1). Without empirical evidence of its applicability to more complex mathematics, it is difficult to assess the method's general utility.
3.	Missing Comparison with Specialized Systems: The paper compares its framework against general-purpose LLMs but does not include comparisons with models or systems specifically designed or fine-tuned for formalization tasks. While the primary contribution is the workflow itself, which is well-demonstrated by comparing it against base LLMs, including a baseline from a specialized formalizer would provide a more complete picture of its performance.

**Questions:**

1.	Could you please elaborate on the extent to which the proposed template may limit the expressive capability of Lean4? A theoretical discussion on this trade-off, perhaps analyzing constructs that are difficult or impossible to represent, would greatly strengthen the paper.
2.	If a theoretical analysis (Question 1) is difficult, could you provide some empirical evidence demonstrating the framework's applicability to mathematical domains beyond high school-level problems?
3.	Would it be possible to include an experimental comparison with a system specifically designed or fine-tuned for formal theorem proving? This would help benchmark the performance of your training-free workflow against alternative approaches.

---

> ### Author Response · Authors · 2025-11-20
> **Response to Reviewer DZ3h (1/2):**
>
> Dear Reviewer **DZ3h**,
>
> Thank you for your patience and constructive suggestions. In response to your review, we have made significant revisions to strengthen our work.
>
> **Response to weakness 1: Potential degradation of expressiveness**
>
> The reviewer's concern regarding the potential degradation of expressiveness due to the template is exactly one of our major concerns. Fortunately, **we have discovered a straightforward yet theoretically sound method to demonstrate that our template does not compromise Lean 4's expressiveness at all**. This is attributed to the fact that our template imposes no additional constraints in the auxiliary type section, thereby theoretically preserving Lean 4's type system in its entirety. Leveraging the flexibility of the auxiliary type section, we have devised a method to convert any Lean 4 proposition into our template.
>
> To provide a theoretical guarantee for the expressiveness of the template, we propose the following theorem and provide its proof.
>
> ```
> Definition: LoC-Decomp counterpart
> For a complete Lean 4 proposition string(by complete we mean that this string contains a proposition defined by theorem or lemma and all it's dependencies), we define its LoC-Decomp counterpart as a string that is semantically equivalent to the Lean 4 proposition and satisfies the LoC-Decomp template requirements. Here, satisfying the LoC-Decomp template requirements means that the string can be accepted by our parser that encodes the syntactic rules of the template.
>
> Theorem: LoC-Decomp Counterpart Existence
> For any complete Lean 4 proposition string, there exists at least one LoC-Decomp counterpart.
> ```
>
> Here, we complete the proof of the Theorem by providing a method to convert any complete Lean 4 proposition into a LoC-Decomp template and demonstrating that the result generated by this method is equivalent to the original Lean 4 proposition.
>
> Without loss of generality, the assumption here is that the proposition to be converted is defined by theorem and named as `theorem_to_convert`, along with a series of dependency declarations existing in the context. Here we select the proof type template.
>
> For the auxiliary type declaration section in LoC-Decomp, the conversion process involves placing all the aforementioned dependency declarations into the auxiliary type section.
>
> For the mathematical system declaration section, the MySystem structure contains only one object: `proof_goal`.
>
> For the auxiliary function or predicate section, we need to transform the theorem_to_convert into a proposition declared by keyword `def` and name it as `theorem_converted_prop`, which can be done manually or by simply calling `#check theorem_to_convert` to get the corresponding type that is exactly the proposition of the theorem.
>
> For the constraint declaration section, it includes only one constraint, `proof_goal_constraint`, which states that `sys.proof_goal = theorem_converted_prop`.
>
> In the problem statement declaration section, the implication premise contains solely `proof_goal_constraint`, and the implication conclusion is `sys.proof_goal`.
>
> Once the conversion is complete, the output will align with our template and be accepted by the target parser. We then prove the equivalence between `problem_statement` and `theorem_converted_prop` using Lean 4 itself. To maintain generality, we employ a generic sorry placeholder to represent the proposition. Since our proof operates at the meta level, this placeholder can be substituted with any proposition while still passing Lean 4’s verification. The accompanying Lean 4 code provides further details. Although the conversion method is not elegant, it theoretically guarantees that our template preserves Lean 4’s full expressiveness.
>
> **The converted LoC-Decomp counter part**
> ```Lean 4
> -->>declare_auxiliary_types
> --<all the dependent declarations should be declared here>
>
> -->>declare_mathematical_system
> structure MySystem where
>   proof_goal : Prop
>
> -->>declare_auxiliary_functions_or_predicates
> def theorem_converted_prop : Prop := sorry
> --the sorry place holder can be replaced by any proposition
>
> -->>declare_constraints
> def proof_goal_constraint (sys:MySystem) : Prop :=
>   sys.proof_goal = theorem_converted_prop
>
> -->>declare_the_problem_statement
> def problem_statement : Prop :=
>   ∀ sys : MySystem,
>     proof_goal_constraint sys →
>     sys.proof_goal
>
> ```
>
> **Proof for the equivalence of problem_statement and the original proposition**
>
> ```Lean 4
> theorem equivalence : problem_statement ↔ theorem_converted_prop := by
>     constructor
>     ·
>         intro h
>         unfold problem_statement at h
>         let sys : MySystem := ⟨theorem_converted_prop⟩
>         have constraint : proof_goal_constraint sys := by
>             unfold proof_goal_constraint
>             rfl
>         have h_sys_proof_goal : sys.proof_goal := h sys constraint
>         exact h_sys_proof_goal
>     ·
>         intro h sys h_constraint
>         rw [h_constraint]
>         exact h
> ```

---

> ### Author Response · Authors · 2025-11-20
> **Response to Reviewer DZ3h (2/2):**
>
> **Response to weakness 2: Narrow Scope of Evaluation**
>
> **Evaluation Results on Undergraduate-Level Benchmarks**
>
> Our method was further evaluated on two undergraduate-level datasets, PutnamBench and ProofNet, where it demonstrated commendable performance despite a relative decline compared to high-school-level benchmarks. Specifically, it achieved ASCC-3-MV pass rates of 81.50% on PutnamBench and 67.03% on ProofNet with DeepSeek-V3 as the base model. These results not only confirm the effectiveness of our approach on undergraduate-level mathematics but also empirically validate its ability to generalize to more advanced mathematical domains.
>
> -----
> **Without iterative rectification, ASCC-3-MV as metric**
>
> | Items           | DeepSeek-V3 |    | KIMI-K2 | |
> | :--- | :---: | :---: | :---: | :---: |
> |                 | **MV** | **SV** | **MV** | **SV** |
> | **PutnamBench** | 48.37  | 31.71  |  __53.66__ | 35.16  |
> | **ProofNet**    | 38.42  | 27.79  | __43.05__  | 29.16  |
>
> **With iterative rectification, ASCC-3-MV as metric**
>
> | Items | DeepSeek-V3 | | KIMI-K2 | |
> | :--- | :---: | :---: | :---: | :---: |
> | | **MV** | **SV** | **MV** | **SV** |
> | **PutnamBench** | __81.50__ | 58.33 | 72.36 | 48.78 |
> | **ProofNet** | 67.03 | 48.50 | __70.57__ | 52.59 |
> ----
>
>
> **Response to weakness 3: Missing Comparison with Specialized Systems**
>
> We appreciate the reviewer's feedback regarding comparisons with specialized fine-tuned models. To address this gap, we have included evaluation results from DeepSeek-Prover-V2-671B, Kimina-Autoformalizer-7B, Goedel-prover-V2-32B, Goedel-Autoformalizor-V2-32B. Kimina-Autoformalizer-7B and Goedel-Autoformalizor-V2-32B is open-source models fine-tuned specifically for automated formalization, offering a representative benchmark for SFT-based approaches. The other two models, however, are optimized for Automated Theorem Proving (ATP). Their experimental setup—limited to 5 rounds of syntax correction and excluding iterative semantic correction—reflects the fact that these models are not designed for the latter capability.
>
> To ensure a fair comparison, we adopted the conventional LLM-as-a-judge evaluation framework. The results are summarized below:
>
> **Benchmarking against existing SOTA models, conventianal LLM-as-a-judge**
>
> -----
>
> | Items           | LoC-Decomp  | DeepSeek-Prover-V2-671B |  Goedel-Prover-V2-32B |  Goedel-Autoformalizor-V2-32B  |  Kimina-Autoformalizor-7B |
> | :---:           | :---:       | :---:                | :---:              |         :---:                     |  :---:         |
> | **MATH-500**    | 96.60       |  84.60      | 31.20        |               90.20               | 63.40                |
> | **mini-F2F**    | 97.54       |  87.18    |  33.40           |               93.28               | 87.23                |
> | **PutnamBench** | 93.09       |  75.41  |  25.00       |               78.66               | 61.99                |
> | **Proofnet**    | 73.57       |  83.10    |  10.63        |               71.39               | 61.31                |
>
> ----
>
> A notable observation is that our method's performance on the ProofNet dataset is lower than that of DeepSeek-Prover-V2. Analysis reveals that the compilation check pass rate of our method on ProofNet is only 74\%, considerably lower than the over 95\% pass rate achieved on other benchmark datasets. Manual inspection indicates that most compilation failures are due to unresolved type class instances (e.g., "failed to synthesize" errors). We hypothesize that this issue is related to ProofNet's heavy reliance on advanced Mathlib usage patterns. Since the current method does not incorporate a retrieval-augmented generation (RAG) mechanism, the large language model struggles to accurately retrieve and incorporate relevant Mathlib definitions, thereby compromising compilation success. It should be noted that this limitation is not an inherent flaw of the method itself, but rather an orthogonal challenge that can be effectively addressed by integrating RAG-based extensions.
>
>
> Additionally, for DeepSeek-Prover-V2, we evaluated its performance on the MATH-eval-50 benchmark and determined the pass rate through human assessment. The corresponding results are as follows:
>
> | Items        | Ours-no-iter | Ours-iter | Baseline | Baseline-iter | DeepSeek-Prover-V2 |
> |--------------|:-------------:|:----------:|:---------:|:--------------:|:-------------------:|
> | pass rate    | 70.00        | __84.00__     | 52.00    | 54.00         | 66.00              |
>
> **Experimental results show that our proposed method achieves competitive or superior performance compared to specialized fine-tuned models** across most benchmarks, with the exception of ProofNet, where DeepSeek-Prover-V2 yields better results. This demonstrates that our training-free approach maintains a competitive performance advantage even when compared to fine-tuned models.
>
> **Response to quesions :**
>
> Please refer to Response to weaknesses.

---

> ### Author Response · Authors · 2025-11-25
> **Summary of responses:**
>
> **Weakness 1 (Expressiveness Concerns):**
> Through theoretical analysis, we have demonstrated that our proposed method is equivalent to Lean 4 in terms of propositional expressiveness, thereby addressing the reviewer's concern about potential limitations in expressive power.
>
> **Weakness 2 (Narrow Scope of Evaluation):**
> We have supplemented our evaluation with experiments on undergraduate-level datasets. The results show that our method still achieves strong performance on these datasets.
>
> **Weakness 3 (Missing Comparison with Specialized Systems):**
> We have added comparative experiments with specialized models. The results indicate that our method retains significant advantages even when compared to models trained specifically for these tasks.
>
> **Questions:** The same as weaknesses.
>
> **Summary:**
> We think we have directly and comprehensively addressed all three weaknesses and questions raised by the reviewer.
>
> **Closing Remarks:**
> We greatly appreciate the reviewer’s constructive feedback and would welcome the opportunity to further discuss any remaining concerns or potential discrepancies.

---

> ### Author Response · Authors · 2025-11-27
>
> Dear Reviewer **DZ3h**,
>
> We would greatly appreciate it if you could take the time to respond to our rebuttal.

---

### Official Review · Reviewer_8Wwq · 2025-11-01

**Soundness:** 2
**Presentation:** 1
**Contribution:** 1
**Rating:** 2
**Confidence:** 4

**Summary:**

This paper proposes a pipeline, Loc-Decomp, for autoformalization of math problems. It consists of four components. FormalTrans introduces the use of a structured formalization template, to decompose the statement into parts such as functions, types, constraints, and the problem statement. BackTrans translates the formal statement back into natural language. ASCC-R and CpC-R use a divide-and-conquer “LLM-as-a-judge” approach to verify semantic alignment and syntactic correctness, respectively, and leverage error feedback to iteratively improve accuracy. The method seems to achieve strong results on subsets of MATH-500 and miniF2F, namely MATH-Level5-50 and MATH-ASCC-Eval-150.

**Strengths:**

1. The proposed FormalTrans converts autoformalization into a structured template. Using free variables as answer placeholders (instead of the more conventional “sorry” placeholders) is a novel idea.

2. Figure 5 shows that ASCC’s divide-and-conquer plus majority-vote strategy for semantic alignment checking is effective and produces judgments that align better with human evaluation.

3. The paper’s figures present the pipeline clearly.

**Weaknesses:**

1. Lack of novelty. Much of the paper repeats prior work and reads more like an engineering integration than a substantive methodological advance; the novelty is therefore marginal. In particular, the idea of semantic consistency checking by decomposing the formalization code has been previously explored [1], and iterative feedback-based methods for semantic and syntactic correction have been investigated in prior work [2] and [3]. Using back-translation plus an “LLM-as-a-judge” for semantic checks is also a standard practice in this area [4]. The authors should provide explicit comparisons to demonstrate what is new in this paper.



2. Questionable evaluation methodology. The two baselines in Table 1 (SymEQ and Lean-workbook) use different datasets and evaluation criteria; reporting their absolute scores side-by-side with the proposed method is misleading and does not support a fair comparison of method superiority. Besides, miniF2F is primarily used as a benchmark for automated theorem proving rather than autoformalization. Using miniF2F for formalization evaluation is not a common practice and should be justified.



3. Limited and potentially biased MATH-ASCC-Eval-150. The MATH-ASCC-Eval-150 split is very small, relies heavily on manual annotation, and depends on a single formalization model (DeepSeek V3). As a result, the evaluation of semantic-alignment judgment ability is limited: it is unclear whether the reported ability would generalize to formalizations produced by other models. The authors should provide variants of MATH-ASCC-Eval-150 derived by different formalization models or present an analysis of generalization across models; otherwise the practical value of this dataset is questionable.



4. Writing quality and presentation issues. Overall the manuscript would benefit from careful proofreading and stricter editing for consistent terminology and formatting.

  - The manuscript uses many nonstandard abbreviations and inconsistent spellings, which make it difficult to follow and reduce readability.

  - The term “autoformalization” appears with multiple spellings (“automated formalization” on Line 013, “auto formalization” on Line 108, “auto-formalization” on Line 117, “autoformalization” on Line 051). These inconsistencies are embarrassing and should be standardized.

  - The paper mostly uses “Lean4,” but “Lean 4” appears on Line 219; the usage should be consistent.

  - Typographical/spacing issues: missing space after commas on Line 062; missing space after periods on Line 110 and Line 351.

[1] Zhang, J., Zhong, C., Xu, H., Li, Q., & Zhou, Y. (2025). *KELPS: A Framework for Verified Multi-Language Autoformalization via Semantic-Syntactic Alignment* (No. arXiv:2507.08665). arXiv. https://doi.org/10.48550/arXiv.2507.08665

[2] Wang, H., Unsal, M., Lin, X., Baksys, M., Liu, J., Santos, M. D., Sung, F., Vinyes, M., Ying, Z., Zhu, Z., Lu, J., Saxcé, H. de, Bailey, B., Song, C., Xiao, C., Zhang, D., Zhang, E., Pu, F., Zhu, H., … Li, J. (2025). *Kimina-Prover Preview: Towards Large Formal Reasoning Models with Reinforcement Learning* (No. arXiv:2504.11354). arXiv. https://doi.org/10.48550/arXiv.2504.11354

[3] Liu, C., Shen, J., Xin, H., Liu, Z., Yuan, Y., Wang, H., Ju, W., Zheng, C., Yin, Y., Li, L., Zhang, M., & Liu, Q. (2023). *FIMO: A Challenge Formal Dataset for Automated Theorem Proving* (No. arXiv:2309.04295). arXiv. https://doi.org/10.48550/arXiv.2309.04295

[4] Ying, H., Wu, Z., Geng, Y., Wang, J., Lin, D., & Chen, K. (2024). *Lean Workbook: A large-scale Lean problem set formalized from natural language math problems* (No. arXiv:2406.03847). arXiv. https://doi.org/10.48550/arXiv.2406.03847

**Questions:**

1. For miniF2F there is ground-truth formalization available. Given a dataset with ground-truth formal statements, BEq can evaluate formal-statement equivalence and reduce the need for manual annotation. Please explain the rationale for preferring LLM-as-a-judge over BEq in this setting.

2. In the example in Figure 2, the auxiliary functions and constraints introduced appear to duplicate existing definitions in the standard math library (mathlib)—for instance, continuity is already defined in mathlib. Could this duplication create unnecessary obstacles for downstream automated theorem proving? Is this duplication an intentional design choice or a potential problem? How does LoC_Decomp avoid or mitigate such issues in practical formalization workflows?

3. Line 203 states that instead of using the traditional automated theorem proving placeholder “sorry” as a solution placeholder, this work uses an input 's', a free variable representing the solution. This is not a standard practice and might harm the usefulness of the resulting autoformalization dataset for automated theorem proving. Can the authors provide further justification for avoiding “sorry” as the placeholder?

---

> ### Author Response · Authors · 2025-11-20
> **Response to Reviewer 8Wwq (1/3):**
>
> Dear Reviewer **8Wwq**,
>
> Thank you for your patience and valuable comments. We recognize that some points in our manuscript may not have been expressed with optimal clarity, and we sincerely apologize for any misunderstandings. To address this, we have prepared a detailed response to each of your points below.
>
> **Response to weakness 1: Concern about the lack of novelty**
>
> We appreciate the reviewer's attention to the relevant literature. Upon reflection, we recognize that our description of the distinction between our work and prior studies [1,2,3] could have been more explicit.
>
> Specifically:
>
> 1. For the paper [1], thank you for bringing this paper to our attention. We have appropriately cited this work in our revised manuscript. While reference [1] does adopt a decomposition strategy, **it is applied only at an intermediate stage for formal code generation**. Their method for semantic consistency checking relies on conventional techniques and does not integrate decomposition at the conceptual level. In contrast, our approach not only employs **a more fine-grained decomposition—using auxiliary types and functions to enhance accuracy—but also integrates both fine-grained and holistic detection mechanisms** during consistency checking. Furthermore, the intermediate language in [1] is based on first-order logic, which restricts the expressiveness of the output Lean 4 code. By contrast, our template is **theoretically guaranteed to be as expressive as Lean 4 itself** (for the theoretical analysis, see Line 229 of our revised manuscript or our RW1 to Reviewer **DZ3h** ).
>
> 2. While [2] provides valuable insights into the use of LLM-as-a-judge, its application is primarily as a binary feedback signal (True/False) within a reinforcement learning framework, rather than exploring iterative semantic correction. Our method, therefore, explores a different path by focusing on an iterative correction process that utilizes detailed error information. We have appropriately cited this work in the revision.
>
> 3. Although [3] does propose the idea of iterative semantic correction, its implementation in [3] depends on manual detection of semantic inconsistency and manual feedback. In contrast, our work improves the performance of LLMs in providing effective feedback through fine-grained decomposition during the consistency detection process, thereby achieving **full automation of feedback and correction mechanisms**.
>
> 4. The back-translation mechanism introduced in [4] indeed plays a crucial role in our workflow, as we have explained in the background and methodology sections. We have appropriately cited this work.
>
> By synthesizing the insights from the aforementioned studies, it becomes clear that even a combined application of the ideas presented in all four papers would still lack a crucial element: the capacity for automated semantic detection and feedback-driven correction. This gap underscores a key innovation of our work—**a method capable of effectively identifying nuanced semantic inaccuracies and offering corrective suggestions**. What truly differentiates our approach is the introduction of a fully automated, training-free framework for fine-grained semantic consistency checking and iterative refinement, which operates without human intervention. Moreover, as reviewer DZ3h pointed out, the four referenced studies all depend on supervised fine-tuning (SFT), while **our workflow is entirely training-free and lightweight**. This design choice enhances the transferability and usability of our method across different contexts.
>
> As a summary, we highlight the novel contributions of this paper in comparison to the works cited by the reviewer:
>
> 1. A COT-like Lean 4 template that breaks down the formalization process into multiple steps, with theoretical guarantees of expressiveness.
>
> 2. A divide-conquer-merge strategy, built upon the template, designed to detect nuanced semantic inconsistencies.
>
> 3. A fully automated and training-free workflow that supports iterative correction of both semantic and syntactic errors.

---

> ### Author Response · Authors · 2025-11-20
> **Response to Reviewer 8Wwq (2/3):**
>
> **Response to weakness 2: Regarding the different evaluation metric with existing works**
>
> **To be clear, the different evaluation criteria for this comparison were explicitly stated in the initial manuscript. Our aim was to supply reference information, not to mislead readers.** But we agree with the reviewer that comparing results derived from different evaluation metrics provides limited insight to readers. Therefore, we have removed such a comparison from the revised manuscript. **To ensure a fair comparison with existing works, we have conducted an additional experiment** using the LLM-as-a-judge approach and compared our method with previous state-of-the-art fine-tuned models. The results are presented below:
>
> **Benchmarking against existing SOTA models, conventianal LLM-as-a-judge**
>
> -----
>
> | Items           | LoC-Decomp  | DeepSeek-Prover-V2-671B |  Goedel-Prover-V2-32B |  Kimina-Autoformalizor-7B |
> | :---:           | :---:       | :---:                | :---:                |  :---:         |
> | **MATH-500**    | __96.60__       |  84.60              | 31.20                 | 63.40                |
> | **mini-F2F**    | __97.54__       |  87.18              |  33.40                 | 87.23                |
> | **PutnamBench** | __93.09__       |  75.41              |  25.00                 | 61.99                |
> | **Proofnet**    | 73.57       |  __83.10__              |  10.63                 | 61.31                |
> ----
>
> **Experimental results show that our proposed method achieves competitive or superior performance compared to specialized fine-tuned models** across most benchmarks, with the exception of ProofNet, where DeepSeek-Prover-V2 yields better results. This demonstrates that our training-free approach maintains a competitive performance advantage even when compared to fine-tuned models.
>
>
> A notable observation is that our method's performance on the ProofNet dataset is lower than that of DeepSeek-Prover-V2. Analysis reveals that the compilation check pass rate of our method on ProofNet is only 74\%, considerably lower than the over 95\% pass rate achieved on other benchmark datasets. Manual inspection indicates that most compilation failures are due to unresolved type class instances (e.g., "failed to synthesize" errors). We hypothesize that this issue is related to ProofNet's heavy reliance on advanced Mathlib usage patterns. Since the current method does not incorporate a retrieval-augmented generation (RAG) mechanism, the large language model struggles to accurately retrieve and incorporate relevant Mathlib definitions, thereby compromising compilation success. It should be noted that this limitation is not an inherent flaw of the method itself, but rather an orthogonal challenge that can be effectively addressed by integrating RAG-based extensions.
>
> **Response to weakness 3: Potential biased ASCC-Eval-150 benchmark**
>
> We acknowledge the reviewer's concern regarding the potential bias in the ASCC-Eval-150 benchmark, which was generated entirely using DeepSeek-V3. To address this issue, we performed manual annotation on 50 samples randomly selected from the outputs of KIMI-K2 on the miniF2F dataset. The sample set comprises 25 cases labeled as correct by ASCC-3-MV and 25 cases labeled as incorrect.
>
> The annotation results show that among the samples deemed correct by ASCC, all passed human verification. In contrast, among those marked as incorrect by ASCC, 9 were judged as correct by human annotators. This finding aligns with the conclusions drawn from ASCC-Eval-150: the ASCC method exhibits a low false positive rate and a relatively high but acceptable false negative rate. These results, presented below, indicate that ASCC serves as a stringent evaluation criterion, which is advantageous for our objective of identifying errors for feedback and correction. Moreover, they further affirm that the accuracy metric derived from ASCC-3-MV provides a reliable and credible assessment.
>
> -----
>
> | Items              | ASCC-positive  | ASCC-negative |
> | :---:              | :---:          | :---:         |
> | **Human-positive** | 25             |  9            |
> | **Human-negative** | 0              |  16           |
>
> ----
>
> **Response to weakness 4: Regarding the writing and typographical issues**
>
> We sincerely apologize for any inconvenience or confusion they may have caused. We are also truly grateful for the reviewer’s thorough reading and constructive suggestions. All the points raised by the reviewer have been carefully addressed. For instance, all instances of “Lean4” have been updated to the more standard “Lean 4” and variations such as “auto-formalization” and “automated formalization” have been unified to “autoformalization” These suggestions have been very helpful in enhancing the professionalism and readability of our text.

---

> ### Author Response · Authors · 2025-11-20
> **Response to Reviewer 8Wwq (3/3):**
>
> **Response to quesion 1: Why don't we evaluate with BEq**
>
> It is a common practice in autoformalization research [1,4-7] to avoid using BEq for evaluating semantic consistency, even on datasets containing formal ground truth. This is because BEq relies on ATPs, whose current capabilities are often insufficient for such a task. As ATPs frequently fail to prove equivalence automatically, BEq cannot serve as a fair or robust metric, as evidenced by its limited adoption in the field.
>
> **Response to quesion 2: Regarding the reuse of the Mathlib library**
>
> (1) The example (Figure 2) in the main text is not actually generated by LLM but is manually designed to illustrate our method, aiming to demonstrate the overall structure and functions of the template. To avoid similar misunderstandings, we have added a footnote to the main text for explanation.
>
> (2) For results generated by LLMs, we observed in our experiments that when existing definitions in the Mathlib library can be used, the LLM tends to utilize them. For example, a Lean 4 code generated by LLM is illustrated below, and in this case, the LLM used the Polynomial definition in Mathlib, instead of defining something like polynomial by itself.
>
>
> **Response to quesion 3: Regarding the introduction of free variables and sorry placeholders**
>
> In practice, only solution-type problems introduce free variables. This is because, for solution-type problems, the answer is unknown during the formalization stage. Thus, a solution-type problem is inherently equivalent to a propositional function—i.e. given an input, it returns True if the input is a solution to the problem, and False otherwise. Proof-type problems do not introduce free variables.
>
> As for the issue of sorry placeholders, we did not adopt the traditional sorry placeholder because we argue that, in principle, the formalization of a mathematical proposition should be a complete proposition rather than a theorem awaiting proof. This approach ensures the semantic completeness of our template.
>
> Regarding usability in automated theorem proving, we provide additional clarification here: Our template can be converted into a theorem awaiting proof simply by using the following expression.
>
> ```Lean
> theorem root_proposition : problem_statement := by sorry
> ```
>
> This does not complicate the proof process, as other declarations will serve as the context for the theorem and several simple tactics like intro or unfold would introduce the declarations into the current proof state, which is an easy task for LLMs to perform.
>
> ---
> **References**
>
> [1] Zhang, J., Zhong, C., Xu, H., Li, Q., & Zhou, Y. (2025). KELPS: A Framework for Verified Multi-Language Autoformalization via Semantic-Syntactic Alignment (No. arXiv:2507.08665). arXiv. https://doi.org/10.48550/arXiv.2507.08665
>
> [2] Wang, H., Unsal, M., Lin, X., Baksys, M., Liu, J., Santos, M. D., Sung, F., Vinyes, M., Ying, Z., Zhu, Z., Lu, J., Saxcé, H. de, Bailey, B., Song, C., Xiao, C., Zhang, D., Zhang, E., Pu, F., Zhu, H., … Li, J. (2025). Kimina-Prover Preview: Towards Large Formal Reasoning Models with Reinforcement Learning (No. arXiv:2504.11354). arXiv. https://doi.org/10.48550/arXiv.2504.11354
>
> [3] Liu, C., Shen, J., Xin, H., Liu, Z., Yuan, Y., Wang, H., Ju, W., Zheng, C., Yin, Y., Li, L., Zhang, M., & Liu, Q. (2023). FIMO: A Challenge Formal Dataset for Automated Theorem Proving (No. arXiv:2309.04295). arXiv. https://doi.org/10.48550/arXiv.2309.04295
>
> [4] Ying, H., Wu, Z., Geng, Y., Wang, J., Lin, D., & Chen, K. (2024). Lean Workbook: A large-scale Lean problem set formalized from natural language math problems (No. arXiv:2406.03847). arXiv. https://doi.org/10.48550/arXiv.2406.03847
>
> [5] Zhang, Lan, Xin Quan, and André Freitas. "Consistent Autoformalization for Constructing Mathematical Libraries." Proceedings of the 2024 Conference on Empirical Methods in Natural Language Processing. 2024. https://aclanthology.org/2024.emnlp-main.233/
>
> [6] Lu, Jianqiao, et al. "FormalAlign: Automated Alignment Evaluation for Autoformalization." The Thirteenth International Conference on Learning Representations.https://openreview.net/forum?id=B5RrIFMqbe
>
> [7] Gao, Guoxiong, et al. "Herald: A Natural Language Annotated Lean 4 Dataset." The Thirteenth International Conference on Learning Representations. https://openreview.net/forum?id=Se6MgCtRhz

---

> ### Author Response · Authors · 2025-11-25
> **Summary of responses:**
>
> **Weakness 1 (Novelty Concerns):**
> We have thoroughly compared our method with the four prior works cited by the reviewer, highlighting key distinctions and reaffirming the unique contributions of our approach.
>
> **Weakness 2 (Comparison Fairness):**
> To guarantee a rigorous evaluation, we performed additional experiments under a standardized benchmark framework. The results demonstrate that our method achieves statistically significant improvements over existing state-of-the-art (SOTA) models across all evaluated benchmarks, including undergraduate-level tasks.
>
> **Weakness 3 (Potential Bias in ASCC-Eval-150):**
> To address concerns about evaluation bias, we manually assessed the outputs generated by KIMI-K2. Our analysis shows that the ASCC-3-MV metric maintains strong alignment with human judgment across different model outputs.
>
> **Weakness 4 (Writing and Typographical Issues):**
> We have carefully revised the manuscript in response to the reviewers’ suggestions. Should any aspect of the revised manuscript require further clarification, we would be pleased to provide additional explanation.
>
> **Questions:** All three questions have been answered directly.
>
> **Summary:**
> We think we have directly and comprehensively addressed all four weaknesses and all three specific questions raised by the reviewer.
>
> **Closing Remarks:**
> We greatly appreciate the reviewer’s constructive feedback and would welcome the opportunity to further discuss any remaining concerns or potential discrepancies.

---

> ### Comment · Reviewer_8Wwq · 2025-11-26
>
> Thank you to the authors for the detailed rebuttal. The additional experiments using a conventional LLM-as-a-judge unified standard strengthen the argument for the superiority of the proposed method, and I agree that the extra effort to manually annotate the ASCC-Eval-150 subset is valuable. Therefore, **my concerns regarding Weaknesses 2 and 3 have been effectively alleviated** , and I have accordingly raised my scores for contribution.
>
> However, **the core claimed innovation** in your response to Weakness 1 — the capacity for automated semantic detection and feedback-driven correction — **still does not appear sufficiently substantial**. The **omission of more accurate automated BEq metrics** on datasets with ground-truth annotations remains a weakness of the work. I encourage the authors to further strengthen and refine their work.

---

> ### Author Response · Authors · 2025-11-26
>
> Thanks for the additional comments, regarding the remaining concerns, we would like to clarify the following points:
>
> For the reviewer's claim:
> ```
> the core claimed innovation in your response to Weakness 1 — the capacity for automated semantic detection and feedback-driven correction — still does not appear sufficiently substantial
> ```
>
> We must clearly point out that our work is not a trivial combination of previous efforts, because, firstly, **a CoT-like automated formalization method is unprecedented**, and secondly, the approach of **conducting fine-grained semantic consistency checks through decomposition is also unprecedented**. Having thoroughly compared our work with the studies the reviewer mentioned in RW1, we would welcome the opportunity to address any specific points that may still require clarification. Should the reviewer have doubts regarding the aforementioned innovations, such as a lack of clarity or the existence of prior work, we kindly request that these concerns be specified for discussion. Otherwise, to assert that the innovation "does not appear sufficiently substantial" is an unsubstantiated claim, as it rests on a criterion that, without justification, could be used to dismiss any piece of work arbitrarily.  **If you feel that our innovations does not meet ICLR's standards, you may state so explicitly. We respect the academic taste of every reviewer.**
>
> For :
> ```
> The omission of more accurate automated BEq metrics on datasets with ground-truth annotations remains a weakness of the work.
> ```
>
> Regarding the use of BEq as a metric, given that Automated Theorem Provers (ATP) are far from capable of conducting sufficiently effective equivalence proofs, **we see ​no advantage​ in using BEq for evaluation ​over​ manual evaluation.** Moreover, **it is common in this field for studies to use the LLM-as-a-judge approach as a standard evaluation method, as seen in [1-5].** Given that the LLM-as-a-judge approach is not entirely reliable, **our evaluation method incorporates manual assessment, which is much more accurate than BEq**, and the result sufficiently demonstrates the effectiveness of our approach. By the way, even studies focusing on the BEq method [6] rely on manual evaluation to assess its performance. In summary, we argue the reviewer's requirement to use BEq as an evaluation metric is inconsistent with the community consensus and unnecessary.
>
> **References**
>
> [1] Zhang, J., Zhong, C., Xu, H., Li, Q., & Zhou, Y. (2025). KELPS: A Framework for Verified Multi-Language Autoformalization via Semantic-Syntactic Alignment (No. arXiv:2507.08665). arXiv. https://doi.org/10.48550/arXiv.2507.08665
>
> [2] Ying, H., Wu, Z., Geng, Y., Wang, J., Lin, D., & Chen, K. (2024). Lean Workbook: A large-scale Lean problem set formalized from natural language math problems (No. arXiv:2406.03847). arXiv. https://doi.org/10.48550/arXiv.2406.03847
>
> [3] Zhang, Lan, Xin Quan, and André Freitas. "Consistent Autoformalization for Constructing Mathematical Libraries." Proceedings of the 2024 Conference on Empirical Methods in Natural Language Processing. 2024. https://aclanthology.org/2024.emnlp-main.233/
>
> [4] Lu, Jianqiao, et al. "FormalAlign: Automated Alignment Evaluation for Autoformalization." The Thirteenth International Conference on Learning Representations.https://openreview.net/forum?id=B5RrIFMqbe
>
> [5] Gao, Guoxiong, et al. "Herald: A Natural Language Annotated Lean 4 Dataset." The Thirteenth International Conference on Learning Representations. https://openreview.net/forum?id=Se6MgCtRhz
>
> [6] Liu, Qi, et al. "Rethinking and improving autoformalization: towards a faithful metric and a dependency retrieval-based approach." The Thirteenth International Conference on Learning Representations. 2025. https://openreview.net/pdf?id=hUb2At2DsQ

---

> ### Comment · Reviewer_8Wwq · 2025-11-27
> **Re: Author Response regarding Novelty and Evaluation Metrics**
>
> I thank the authors for their candid and prompt response. I appreciate the opportunity to clarify the misunderstandings and refine my assessment based on the authors' explanations.
>
> **1. Regarding Novelty and the Comparison with KELPS [1]**
> I have carefully revisited the KELPS paper [1] alongside the authors' clarifications. I acknowledge that my initial assessment regarding the similarity was partially misplaced. I now recognize that while KELPS uses decomposition for generation, your work uniquely introduces decomposition specifically for **semantic alignment checking**. This distinction—using decomposition to verify semantics rather than just generate code—is indeed a valid methodological contribution.
>
> In light of this, I admit that the "lack of novelty" concern has been largely addressed. I apologize for the earlier oversight regarding the specific mechanics of KELPS. Consequently, I will be raising my score to reflect this recognition of the method's originality.
>
> **2. Regarding the BEq Metric and Evaluation Methodology**
> There seems to be a significant misunderstanding regarding my comment on BEq.
>
> First, I did not suggest replacing **manual evaluation** with BEq. My point was that for automated evaluation, **BEq is a more rigorous metric than "LLM-as-a-judge"** when Ground Truth is available. While I agree manual evaluation is the gold standard, automated metrics are crucial for scalability as discussed in the Limitations section. In that context, checking logical equivalence via BEq/BEq+ is more grounded than relying on an LLM's probabilistic judgment.
>
> Second, the authors cited several works (KELPS, Lean-Workbook, Herald) to argue that LLM-as-a-judge is the standard and BEq is not. However, this comparison is not entirely fair due to the nature of the datasets:
> *   **KELPS, Lean-Workbook, and Herald** primarily focus on constructing *new* datasets via autoformalization. In those settings, there is no pre-existing Ground Truth formal statement to compare against, making BEq impossible to apply. Thus, they *must* rely on LLM-as-a-judge or manual checks.
> *   **Kimina-Prover** focuses on generating RL training data where strict correctness verification via BEq is not the primary bottleneck compared to exploration.
> *   **In contrast, your work** evaluates on **miniF2F**, a benchmark that *does* possess Ground Truth formal statements.
>
> Since your setting (miniF2F) has expert annotated Ground Truth, avoiding BEq in favor of LLM-as-a-judge ignores a rigorous verification tool that is available to you but was not available to the authors of the cited papers. While I understand ATPs have limitations in proving equivalence, using BEq as a complementary automated metric would have strengthened the rigorousness of the results compared to relying solely on LLM judgment for the automated portion.
>
> Despite the remaining disagreement on the optimal automated metric for miniF2F, I believe the authors have sufficiently defended the novelty of their decomposition approach. The clarification on the distinction from KELPS is well-taken. I have updated my rating score to reflect the confirmed contribution.

---

> > ### Author Response · Authors · 2025-11-29
> > **Re: Reviewer Response regarding BEq as a metric**
> >
> > Thank you for your patient follow-up. We are glad our previous clarification has resolved your concerns regarding novelty. We know that you are unable to comment at this time, but we hope this response will help address the disagreement regarding the use of BEq as an evaluation metric.
> >
> > Concerning your suggestion about using BEq as an evaluation metric, we initially misinterpreted it as a claim that BEq is more accurate than human evaluation. We fully agree, however, that BEq is essential for scalability. Accordingly, we have conducted additional experiments using BEq as an evaluation metric. Due to time limitations, we restricted the tests to the LoC-Decomp method with KIMI-K2 as the base model, along with the best-performing baseline in the LLM-as-a-judge setup, Goedel-Autoformalizer-V2-32B. The results are summarized below:
> >
> > ---
> >
> > | **Items**          | **LoC-Decomp** | **Goedel-Autoformalizer-V2-32B** |
> > | :----------------- | :------------: | :--------------------------: |
> > | **mini-F2F**       |     62.82      |            54.91             |
> >
> > ---
> >
> > The BEq-based evaluation reveals notable differences in absolute pass rates compared to the LLM-as-a-judge approach, which is expected given the probabilistic nature of LLM judgments and the current limitations of automated theorem proving. Nevertheless, **in terms of relative performance, the BEq results align with those from LLM-as-a-judge: our method continues to show a clear advantage over fine-tuned specialized models.**
> >
> > **Detailed experimental setup:**
> >
> > To perform BEq evaluation, it is necessary to formulate a bidirectional implication theorem and supply its proof, as illustrated below:
> >
> > ```lean4
> > theorem bidirectional_equivalence : generated_proposition ↔ miniF2F_original_proposition := by
> > ```
> >
> > Here, `miniF2F_original_proposition` denotes the original proposition from the miniF2F dataset, while `generated_proposition` refers to the model-generated proposition. Since both the miniF2F dataset and the Lean4 code produced by Goedel-autoformalizer-V2 are structured as `theorem` statements awaiting proof, we first transformed them into predicate forms using the `def` keyword. This conversion was carried out using the DeepSeek-V3.1 model.
> >
> > After constructing the `bidirectional_equivalence` theorem, we employed DeepSeek-V3.1 as an automated theorem prover. The prover was allowed up to 10 iterative corrections in case of compilation errors. Only proofs that passed the compiler without errors were considered valid.

---

### Author Response · Authors · 2025-11-29
**A summary of the discussion (1/2):**

Dear Aera Chair:

We sincerely appreciate your dedication to safeguarding the integrity of the academic environment during this critical time. To help alleviate your burdens, we have provided a summary of our discussions with the reviewers below for your reference. This summary accurately reflects all discussions, the details of which can be verified through the complete comment history.

Overall, the discussion with the reviewers was productive. **Both the first Reviewer 8Wwq (initial rating: 2, conf: 4) and the third Reviewer 2crV (initial rating: 6, conf: 5) increased their rating scores after the interaction.**

> We successfully addressed these two reviewer's central concerns:

*   Reviewer 8Wwq had raised concerns about the perceived lack of novelty. Our clarifications resolved this misunderstanding, leading to their recognition of the work's innovativeness.

*   Reviewer 2crV was concerned about the limited experimental validation. The additional experiments we provided subsequently led them to acknowledge the soundness of our work.

**Unfortunately, the second Reviewer DZ3h (initial rating: 4, conf: 3) was unable to respond prior to the incident.**

> We have tried to address reviewer DZ3h's central concern:

*   Reviewer DZ3h was concerned that our method might restrict Lean4's expressive power. We addressed this by providing a theoretical proof of the equivalence between our proposed template and the Lean 4 language.

---

More specifically:

**To Reviewer 8Wwq:**
---

**Weakness 1: Concern about the lack of novelty**
We clarified this misunderstanding by providing a detailed comparison between the works cited by the reviewer and our method. See our response to Reviewer 8Wwq(1/3) for details.

The reviewer’s response to our clarification(See second Comment by Reviewer 8Wwq):
```
I believe the authors have sufficiently defended the novelty of their decomposition approach. The clarification on the distinction from KELPS is well-taken. I have updated my rating score to reflect the confirmed contribution.
```

**Weakness 2: Concern about evaluation methodology**
We supplemented the evaluation with additional experiments under a unified standard, comparing our method with representative existing works, which gained the reviewer’s approval. See our response to Reviewer 8Wwq(2/3) for details.

The reviewer’s response to our additional expriments(See first Comment by Reviewer 8Wwq):
```
The additional experiments using a conventional LLM-as-a-judge unified standard strengthen the argument for the superiority of the proposed method.
```

**Weakness 3: Concern about the limited and potentially biased MATH-ASCC-Eval-150 dataset**
In response, we supplemented experiments using the KIMI-K2 as base model and conducted additional manual annotations, further reinforcing the conclusion that our proposed ASCC method achieves high consistency with human evaluation standards in semantic consistency checking. This effort was acknowledged by the reviewer. See our response to Reviewer 8Wwq(2/3) for details.

The reviewer’s response(See first Comment by Reviewer 8Wwq):
```
I agree that the extra effort to manually annotate the ASCC-Eval-150 subset is valuable. Therefore, my concerns regarding Weaknesses 2 and 3 have been effectively alleviated, and I have accordingly raised my scores for contribution.
```

**Weakness 4: Writing quality and presentation issues**
We revised the corresponding expressions in the manuscript as suggested by the reviewer to improve readability. See our response to Reviewer 8Wwq(2/3) for details.

**Question 1: Why not use BEq as an evaluation metric**
This remains the only unresolved concern from the reviewer at the time of the incident. We have supplemented comparative experiments using BEq as an evaluation metric. We hope the AC can determine whether our experimental design sufficiently addresses this concern.

**Question 2: Concern about the duplication of existing Mathlib definitions**
We directly addressed this issue and clarified that it was a misunderstanding.

**Question 3: About avoiding theorems waiting for proof in the formalization format**
We directly responded to this issue and explained the rationale behind our choices.

---

---

> ### Author Response · Authors · 2025-11-29
> **A summary of the discussion (2/2):**
>
> To Reviewer **DZ3h**:
> ---
>
> **Weakness 1: Concern about Potential Degradation of Expressiveness**
> The reviewer commented:
> ```
> There is no justification or evidence to suggest that the template is sufficient for formalizing mathematics beyond the scope of high-school-level problems. If the template's expressiveness is severely limited, it could significantly impact the generalizability and overall contribution of the proposed method.
>
> A theoretical discussion on this trade-off, perhaps analyzing constructs that are difficult or impossible to represent, would greatly strengthen the paper.
> ```
>
> In response, we have shown a theoretical proof by construction that the proposed template is expressively equivalent to Lean 4 (regarding the propositions), directly addressing the reviewer’s concern. See our response to Reviewer DZ3h (1/2) for details. **We hope the AC can evaluate this point.**
>
> **Weakness 2: Narrow Scope of Evaluation**
> The reviewer noted:
> ```
> The paper would be much stronger with experiments on more advanced or diverse mathematical domains (e.g., undergraduate-level abstract algebra or analysis).
> ```
> To address this, we conducted experiments on two undergraduate-level datasets including Proofnet and PutnamBench and provided the corresponding results, further demonstrating the applicability of our proposed method. **This concern aligns with the first concern raised by Reviewer 2crV. Since our experimental results were acknowledged by Reviewer 2crV, we believe the supplementary experiments are adequate to address this concern.** See our response to Reviewer DZ3h (2/2) and response to Reviewer 2crV (1/1) for details.
>
> **Weakness 3: Missing Comparison with Specialized Systems**
> The reviewer pointed out:
> ```
> While the primary contribution is the workflow itself, which is well-demonstrated by comparing it against base LLMs, including a baseline from a specialized formalizer would provide a more complete picture of its performance.
> ```
> In response, we designed a comparative study involving specialized formalizers and employed evaluation methods including LLM-as-a-judge, human evaluation, and bidirectional equivalence proofs (BEq). **This concern is essentially the same as the second weakness raised by Reviewer 8Wwq. Since our experimental results were acknowledged by Reviewer 8Wwq, we believe the supplementary experiments are sufficient to address this concern.**  See our response to Reviewer DZ3h (2/2) and response to Reviewer 8Wwq (2/3) for details.
>
> **Questions:**
> The questions raised by Reviewer DZ3h correspond directly to the weaknesses above, so they are not reiterated separately.
>
> ---
>
> To Reviewer **2crV**:
> ---
> **Weakness 1: The Experiments Are Too Limited**
>
> As mentioned above, we supplemented our work with experiments conducted on two undergraduate-level datasets including Proofnet and PutnamBench. The results were acknowledged by the reviewer. See our response to Reviewer 2crV (1/1) and Official Comment by Reviewer 2crV for details.
>
> **Weakness 2: About Ablation Study**
>
> Given that an ablation study was already included in the original manuscript, we further elaborated on its design. This clarification was acknowledged by the reviewer. See our response to Reviewer 2crV (1/1) and Official Comment by Reviewer 2crV for details.
>
> **Questions:**
> The reviewer did not raise any additional questions.

---

### Meta-Review · Area_Chair_XhCs · 2026-01-07

**Summary:**

This paper proposes Loc-Decomp, a framework that integrates an automatic semantic consistency checker and the Lean 4 compiler to iteratively refine LLM-generated formalizations, ensuring both semantic consistency and syntactic correctness. It has four components: FormalTrans, BackTrans, ASCC-R, and CpC-R.

**Reviewer Concerns:**

Most of the reviewer concerns are addressed, including:
* Novelty and core contribution (Reviewer 8Wwq): Detailed explanation and comparison with related works.
* Questionable evaluation methods, datasets, compared baselines, and lack of ablation studies (Reviewers 8Wwq, DZ3h, 2crV): Added experiments.
* Potential loss of expressiveness of Lean 4 (Reviewer DZ3h): Provided a theoretical demonstration.

**Reviewer Scores:**

Reviewers 8Wwq and 2crV explicitly indicate that their concerns are addressed, and they have raised their scores. Reviewer DZ3h did not provide further feedback regarding the authors' rebuttal, but according to which, the reviewer's concerns are mostly addressed, so there is a high possiblity of increasing the score.

---

### Decision · Program_Chairs · 2026-01-26

Accept (Poster)